# Adversarial Surrogate Risk Bounds for Binary Classification

**Natalie Frank**                                                                      *natalief@uw.edu*
*Department of Applied Mathematics*
*University of Washington*

**Reviewed on OpenReview:** *https://openreview.net/forum?id=Bay1cHLk7h*

### Abstract

A central concern in classification is the vulnerability of machine learning models to adversarial attacks. Adversarial training is one of the most popular techniques for training robust classifiers, which involves minimizing an adversarial surrogate risk. Recent work has characterized the conditions under which any sequence minimizing the adversarial surrogate risk also minimizes the adversarial classification risk in the binary setting, a property known as *adversarial consistency*. However, these results do not address the rate at which the adversarial classification risk approaches its optimal value along such a sequence. This paper provides surrogate risk bounds that quantify that convergence rate.

## 1 Introduction

A central concern regarding regarding sophisticated machine learning models is their susceptibility to adversarial attacks. Prior work (Biggio et al., 2013; Szegedy et al., 2013) demonstrated that imperceptible perturbations can degrade the performance of neural nets. As such models are deployed in security-critical applications, including facial recognition (Xu et al., 2022) and medical imaging (Paschali et al., 2018), training robust models remains a key challenge in machine learning.

In the standard classification setting, the *classification risk* is the proportion of incorrectly classified data. Directly minimizing this quantity is a combinatorial optimization problem, so typical machine learning algorithms instead minimize a more tractable *surrogate* risk via gradient-based methods. A surrogate risk is said to be consistent for a given data distribution if every minimizing sequence also minimizes the classification risk for that distribution. Beyond consistency, a central objective is efficiency: minimizing the surrogate risk should translate into a rapid reduction of the classification risk. This rate can be quantified via surrogate risk bounds, which bound the excess classification risk in terms of the excess surrogate risk.

In the standard binary classification setting, consistency and surrogate risk bounds are well-studied topics (Bartlett et al., 2006; Lin, 2004; Steinwart, 2007; Zhang, 2004). A typical approach reduces the problem to a pointwise analysis of the conditional classification and surrogate risks. In contrast, the adversarial setting is less understood. The adversarial classification risk penalizes instances that can be perturbed into the opposite class, while the adversarial surrogate risk computes the worst-case value over an $\epsilon$-ball. The dependence on the value of a function over an $\epsilon$-ball precludes a pointwise decomposition, rendering the classical analysis inapplicable. Frank & Niles-Weed (2024a) characterized the risks that are consistent for all data distributions, and the corresponding losses are referred to as *adversarially consistent*. Unfortunately, no convex loss function can be adversarially consistent for all data distributions (Meunier et al., 2022). On the other hand, Frank (2025) showed that such situations are rather atypical— when the data distribution is absolutely continuous, a surrogate risk is adversarially consistent so long as the adversarial Bayes classifier satisfies a certain notion of uniqueness. While these results characterize consistency, none describe convergence rates.

**Our Contributions:**

- We prove a linear surrogate risk bound for adversarially consistent losses (Theorem 9).

- When the "distribution of optimal attacks" satisfies a bounded noise condition, we prove a linear surrogate risk bound under mild conditions on the loss (Theorem 9).

- We establish a distribution-dependent surrogate risk bound that applies whenever a loss is adversarially consistent for the data distribution (Theorem 11).

Notably, the last result applies to convex loss functions. By prior consistency results (Frank, 2025; Frank & Niles-Weed, 2024a; Meunier et al., 2022), one cannot hope for distribution independent surrogate bounds for non-adversarially consistent losses. This work presents a framework for surrogate risk bounds that applies to any supremum-based risk under mild conditions. A detailed comparison with prior work is provided in Section 7.

## 2 Background and Preliminaries

### 2.1 Surrogate Risks

We study binary classification on $\mathbb{R}^d$ with labels $-1$ and $+1$, where $\mathbb{P}_0$ and $\mathbb{P}_1$ denote the class-conditional distributions. For a measurable set $A$, the classification risk is

$$R(A) = \int 1_{A^C} \, d\mathbb{P}_1 + \int 1_A \, d\mathbb{P}_0,$$

with minimum $R^*$ over all Borel sets. Because the indicator function is nondifferentiable, one instead minimizes a *surrogate risk*

$$R_\phi(f) = \int \phi(f) \, d\mathbb{P}_1 + \int \phi(-f) \, d\mathbb{P}_0,$$

with minimum $R_{\phi,*}$ over all Borel functions. The loss $\phi$ satisfies:

**Assumption 1.** $\phi$ *is continuous, non-increasing, and* $\lim_{\alpha \to \infty} \phi(\alpha) = 0$.

Thresholding $f$ at zero yields the classifier $\{f > 0\}$, whose risk is

$$R(f) = R(\{f > 0\}) = \int 1_{f \leq 0} \, d\mathbb{P}_1 + \int 1_{f > 0} \, d\mathbb{P}_0.$$

It remains to verify that minimizing the surrogate risk $R_\phi$ will also minimize the classification risk $R$.

**Definition 1.** *The loss function $\phi$ is* consistent *for the distribution* $\mathbb{P}_0, \mathbb{P}_1$ *if every minimizing sequence of $R_\phi$ is also a minimizing sequence of $R$. The loss function $\phi$ is* consistent *if it is consistent for all distributions.*

Prior work establishes conditions under which many common loss functions are consistent. For convex $\phi$, consistency occurs iff $\phi$ is differentiable at 0 and $\phi'(0) < 0$ (Bartlett et al., 2006, Theorem 2). Frank & Niles-Weed (2024a, Proposition 3) show that consistency holds if $\inf_\alpha \frac{1}{2}(\phi(\alpha) + \phi(-\alpha)) < \phi(0)$, which is satisfied by losses such as the $\rho$-margin loss $\phi_\rho(\alpha) = \min(1, \max(1 - \alpha/\rho, 0))$ and the shifted sigmoid loss $\phi_\tau(\alpha) = 1/(1 + \exp(\alpha - \tau))$, $\tau > 0$. However, a convex loss $\phi$ cannot satisfy this inequality:

$$\frac{1}{2}(\phi(\alpha) + \phi(-\alpha)) \geq \phi\left(\frac{1}{2}\alpha + \frac{1}{2} \cdot -\alpha\right) = \phi(0). \tag{1}$$

### 2.2 Surrogate Risk Bounds

In addition to consistency, quantifying convergence rates is a key concern. Specifically, prior work (Bartlett et al., 2006; Zhang, 2004) establishes *surrogate risk bounds* of the form $\Psi(R(f) - R_*) \leq R_\phi(f) - R_{\phi,*}$ for some function $\Psi$, linking excess classification risk to excess surrogate risk. These bounds involve pointwise minima of the *conditional* classification and surrogate risks.

Let $\mathbb{P} = \mathbb{P}_0 + \mathbb{P}_1$ and $\eta(\mathbf{x}) = d\mathbb{P}_1/d\mathbb{P}$. An equivalent formulation of the classification risk is

$$R(f) = \int C(\eta(\mathbf{x}), f(\mathbf{x}))d\mathbb{P}(\mathbf{x}) \tag{2}$$

where $C(\eta, \alpha) = \eta\mathbf{1}_{\alpha \leq 0} + (1 - \eta)\mathbf{1}_{\alpha > 0}$, with minimal conditional risk

$$C^*(\eta) = \inf_\alpha C(\eta, \alpha) = \min(\eta, 1 - \eta), \tag{3}$$

and thus the minimal classification risk is $R_* = \int C^*(\eta(\mathbf{x}))d\mathbb{P}(\mathbf{x})$. Analogously, the surrogate risk in terms of $\eta$ and $\mathbb{P}$ is

$$R_\phi(f) = \int C_\phi(\eta(\mathbf{x}), f(\mathbf{x}))d\mathbb{P}, \quad C_\phi(\eta, \alpha) = \eta\phi(\alpha) + (1 - \eta)\phi(-\alpha) \tag{4}$$

and the minimal surrogate risk is $R_{\phi,*} = \int C_\phi^*(\eta(\mathbf{x}))d\mathbb{P}(\mathbf{x})$ with the minimal conditional risk $C_\phi^*(\eta)$ defined by

$$C_\phi^*(\eta) = \inf_\alpha C_\phi(\eta, \alpha). \tag{5}$$

Prior work on consistency typically establishes surrogate risk bounds via pointwise analysis of the conditional risks, relating the excess conditional surrogate risk $C_\phi(\eta, \alpha) - C_\phi^*(\eta)$ to the excess conditional classification risk $C(\eta, \alpha) - C^*(\eta)$.

The consistency of $\phi$ can be fully characterized by the properties of the function $C_\phi^*(\eta)$.

**Theorem 1.** *A loss $\phi$ is consistent iff $C_\phi^*(\eta) < \phi(0)$ for all $\eta \neq 1/2$.*

Surprisingly, this criterion has not appeared in prior work. See Appendix A for a proof. In terms of the function $C_\phi^*$, Frank & Niles-Weed (2024a, Proposition 3) states that any loss $\phi$ with $C_\phi^*(1/2) < \phi(0)$ is consistent. The function $C_\phi^*$ is a key component of surrogate risk bounds from prior work. Specifically, Bartlett et al. (2006) shows:

**Theorem 2** (Tewari & Bartlett (2007))**.** *Let $\phi$ be any loss satisfying Assumption 1 with $C_\phi^*(1/2) = \phi(0)$ and define*

$$\Psi(\theta) = \phi(0) - C_\phi^*\left(\frac{1 + \theta}{2}\right).$$

*Then*

$$\Psi(C(\eta, f) - C^*(\eta)) \leq C_\phi(\eta, f) - C_\phi^*(\eta) \tag{6}$$

*and consequently*

$$\Psi(R(f) - R_*) \leq R_\phi(f) - R_\phi^*. \tag{7}$$

The inequality (7) is a consequence of (6) and Jensen's inequality. Theorem 1 implies that this bound is non-vacuous iff $\phi$ is consistent— compare with Theorem 1. Moreover, (6) yields a distribution-dependent linear surrogate bound when $\eta$ is bounded away from $1/2$. If *Massart's noise condition* (Massart & Nédélec, 2006) holds— namely, there exists a $\alpha \in [0, 1/2]$ for which $|\eta - 1/2| \geq \alpha$ $\mathbb{P}$-a.e., then the distribution admits a linear surrogate bound.

**Proposition 1.** *Let $\eta, \mathbb{P}$ be a distribution that satisfies $|\eta - 1/2| \geq \alpha$ $\mathbb{P}$-a.e. with a constant $\alpha \in [0, 1/2]$, and let $\phi$ be a loss with $\phi(0) > C_\phi^*(1/2 - \alpha)$. Then for all $|\eta - 1/2| \geq \alpha$,*

$$C(\eta, f) - C^*(\eta) \leq \frac{1}{\phi(0) - C_\phi^*(\frac{1}{2} - \alpha)}(C_\phi(\eta, f) - C_\phi^*(\eta)) \tag{8}$$

*and consequently*

$$R(f) - R_* \leq \frac{1}{\phi(0) - C_\phi^*(\frac{1}{2} - \alpha)}(R_\phi(f) - R_{\phi,*}) \tag{9}$$

See Appendix B for a proof of this result. Observe that Theorem 1 guarantees that the linear constant is finite whenever $\alpha \neq 0$ and $\phi$ is consistent. This bound is distribution-independent when $\phi(0) > C_\phi^*(1/2)$ with $\alpha = 0$, and will later be generalized to adversarial risks. Although the constant in Proposition 1 is not optimal, further refinement offers no improvement to our adversarial bounds, so we opt to retain the simpler form.

## 2.3 Adversarial Risks

The adversarial classification risk incurs a penalty of 1 whenever a point $\mathbf{x}$ can be perturbed into the opposite class. This penalty can be expressed in terms of supremums of indicator functions— the adversarial classification risk incurs a penalty of 1 whenever $\sup_{\|\mathbf{x}'-\mathbf{x}\|\leq\epsilon} \mathbf{1}_A(\mathbf{x}') = 1$ or $\sup_{\|\mathbf{x}'-\mathbf{x}\|\leq\epsilon} \mathbf{1}_{A^C}(\mathbf{x}') = 1$. Define

$$S_\epsilon(g)(\mathbf{x}) = \sup_{\|\mathbf{x}-\mathbf{x}'\|\leq\epsilon} g(\mathbf{x}').$$

The adversarial classification and surrogate risks are given respectively by[1]

$$R^\epsilon(A) = \int S_\epsilon(\mathbf{1}_{A^C})d\mathbb{P}_1 + \int S_\epsilon(\mathbf{1}_A)d\mathbb{P}_0, \quad R_\phi^\epsilon(f) = \int S_\epsilon(\phi(f))d\mathbb{P}_1 + \int S_\epsilon(\phi(-f))d\mathbb{P}_0.$$

A minimizer of the adversarial classification risk is called an *adversarial Bayes classifier*. After optimizing the surrogate risk, a classifier is obtained by thresholding the resulting function $f$ at zero. The associated adversarial classification error function $f$ is then

$$R^\epsilon(f) = R^\epsilon(\{f > 0\}) = \int S_\epsilon(\mathbf{1}_{f\leq 0})d\mathbb{P}_1 + \int S_\epsilon(\mathbf{1}_{f>0})d\mathbb{P}_0. \tag{10}$$

Just as in the standard case, one would hope that minimizing the adversarial surrogate risk would minimize the adversarial classification risk.

**Definition 2.** *The loss $\phi$ is* adversarially consistent *for the distribution $\mathbb{P}_0$, $\mathbb{P}_1$ if any minimizing sequence of $R_\phi^\epsilon$ is also a minimizing sequence of $R^\epsilon$. We say that $\phi$ is* adversarially consistent *if it is adversarially consistent for all distributions.*

Theorem 2 of Frank & Niles-Weed (2024a) characterizes the adversarially consistent losses:

**Theorem 3** (Frank & Niles-Weed (2024a))**.** *The loss $\phi$ is adversarially consistent iff $C_\phi^*(1/2) < \phi(0)$.*

Frank & Niles-Weed (2024a, Proposition 3) guarantees that every adversarially consistent loss is also consistent in the standard sense. Unfortunately, (1) shows that no convex loss is adversarially consistent. However, distributions for which consistency fails are atypical: for absolutely continuous $\mathbb{P}$, adversarial consistency holds provided the adversarial Bayes classifier is *unique up to degeneracy.*

**Definition 3.** *Two adversarial Bayes classifiers $A_1$, $A_2$ are* equivalent up to degeneracy *if any set $A$ with $A_1 \cap A_2 \subset A \subset A_1 \cup A_2$ is also an adversarial Bayes classifier. The adversarial Bayes classifier is* unique up to degeneracy *if any two adversarial Bayes classifiers are equivalent up to degeneracy.*

See Figure 1 for an illustration of non-equivalent adversarial Bayes classifiers in a distribution where adversarial consistency fails. Theorem 4 of Frank (2025) relates uniqueness of the adversarial Bayes classifier to the consistency of $\phi$.

**Theorem 4** (Frank (2025))**.** *Let $\phi$ be a loss with $C_\phi^*(1/2) = \phi(0)$ and assume that $\mathbb{P}$ is absolutely continuous with respect to Lebesgue measure. Then $\phi$ is adversarially consistent for the data distribution given by $\mathbb{P}_0$, $\mathbb{P}_1$ iff the adversarial Bayes classifier is unique up to degeneracy.*

Any extension of surrogate risk bounds to the adversarial setting must account for the conditions of Theorems 3 and 4.

---

[1] In order to define the risks $R_\phi^\epsilon$ and $R^\epsilon$, one must argue that $S_\epsilon(g)$ is measurable. Theorem 1 of Frank & Niles-Weed (2024b) proves that whenever $g$ is Borel, $S_\epsilon(g)$ is always measurable with respect to the completion of any Borel measure.

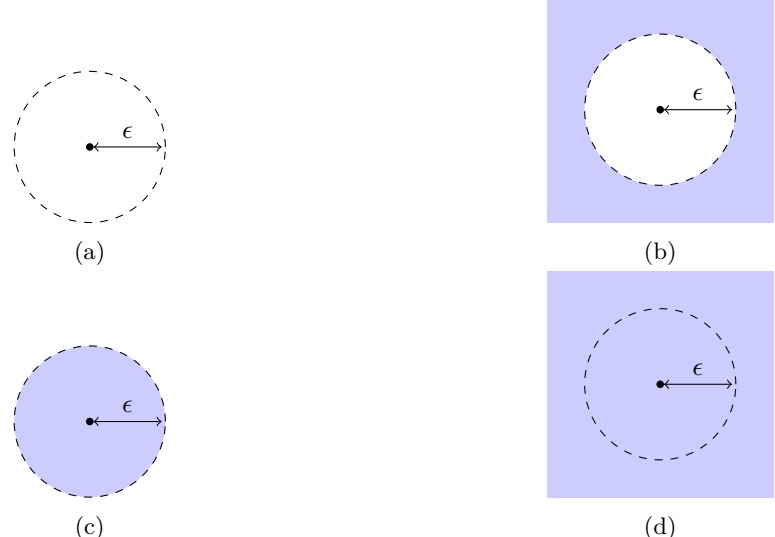

Figure 1: Adversarial Bayes classifiers for the distribution where $\mathbb{P}_0 = \mathbb{P}_1$ are uniform distributions on $\overline{B_\epsilon(\mathbf{0})}$, the counterexample from Meunier et al. (2022). The classifiers in (a) and (b) are equivalent up to degeneracy, as are those in (c) and (d), but the classifiers in (a) and (c) are not. A sequence minimizing $R_\phi^\epsilon$ but not $R^\epsilon$ is provided in (33).

## 2.4 Minimax Theorems

A central tool in analyzing the adversarial consistency of surrogate risks is minimax theorems, which enable a 'pointwise'-style representation of adversarial risks analogous (4). This section reviews the minimax representation for both adversarial classification and surrogate risks, which underlie the bounds in Section 3.

These minimax theorems utilize the $\infty$-Wasserstein ($W_\infty$) metric from optimal transport. Informally, this metric quantifies the smallest radius $\epsilon$ such that the mass of one distribution can be transported to match another without moving any point more than $\epsilon$.

Formally, let $\mathbb{Q}$ and $\mathbb{Q}'$ be finite positive measures with equal total mass. A Borel measure $\gamma$ on $\mathbb{R}^d \times \mathbb{R}^d$ is a coupling between $\mathbb{Q}$ and $\mathbb{Q}'$ if its first marginal is $\mathbb{Q}$ and its second marginal is $\mathbb{Q}'$, or in other words, $\gamma(A \times \mathbb{R}^d) = \mathbb{Q}(A)$ and $\gamma(\mathbb{R}^d \times A) = \mathbb{Q}'(A)$ for all Borel sets $A$. Denote the set of couplings between $\mathbb{Q}$ and $\mathbb{Q}'$ by $\Pi(\mathbb{Q}, \mathbb{Q}')$. Then the $W_\infty$ distance is

$$W_\infty(\mathbb{Q}, \mathbb{Q}') = \inf_{\gamma \in \Pi(\mathbb{Q}, \mathbb{Q}')} \operatorname*{ess\,sup}_{(\mathbf{x}, \mathbf{y}) \sim \gamma} \|\mathbf{x} - \mathbf{y}\|. \tag{11}$$

Theorem 2.6 of Jylhä (2014) proves that the infimum in (11) is always attained. The $\epsilon$-ball around $\mathbb{Q}$ in the $W_\infty$ metric is $\mathcal{B}_\epsilon^\infty(\mathbb{Q}) = \{\mathbb{Q}' : W_\infty(\mathbb{Q}', \mathbb{Q}) \le \epsilon\}$.

The next lemma is a standard observation linking adversarial perturbations to $W_\infty$-balls. We include a proof in Appendix C for completeness; it is a known result and not new to this work (see for instance Matthew Staib (2017, Proposition 3.1)).

**Lemma 1.** *Let $g$ be a Borel function. Let $\gamma$ be a coupling between the measures $\mathbb{Q}$ and $\mathbb{Q}'$ supported on $\Delta_\epsilon = \{(\mathbf{x}, \mathbf{x}') : \|\mathbf{x} - \mathbf{x}'\| \le \epsilon\}$. Then $S_\epsilon(g)(\mathbf{x}) \ge g(\mathbf{x}')$ $\gamma$-a.e. and consequently*

$$\int S_\epsilon(g) d\mathbb{Q} \ge \sup_{\mathbb{Q}' \in \mathcal{B}_\epsilon^\infty(\mathbb{Q})} \int g d\mathbb{Q}'.$$

Applying Lemma 1 to $R^\epsilon$ shows that $\inf_A R^\epsilon(A)$ can be expressed as an inf-sup problem. The minimax theorem of Pydi & Jog (2021) ensures that the order of the inf and sup can be interchanged. Let $C^*(\eta)$ be as defined in (3) and define

$$\bar{R}(\mathbb{P}'_0, \mathbb{P}'_1) = \inf_{A \text{ Borel}} \int \mathbf{1}_{A^C} d\mathbb{P}'_1 + \int \mathbf{1}_A d\mathbb{P}'_0 = \int C^* \left( \frac{d\mathbb{P}'_1}{d(\mathbb{P}'_1 + \mathbb{P}'_0)} \right) d(\mathbb{P}'_0 + \mathbb{P}'_1). \tag{12}$$

**Theorem 5** (Frank (2025)). *Let $\bar{R}$ be as defined in (12). Then*

$$\inf_{A \text{ Borel}} R^\epsilon(A) = \sup_{\substack{\mathbb{P}'_1 \in \mathcal{B}^\infty_\epsilon(\mathbb{P}_1) \\ \mathbb{P}'_0 \in \mathcal{B}^\infty_\epsilon(\mathbb{P}_0)}} \bar{R}(\mathbb{P}'_0, \mathbb{P}'_1).$$

*with equality attained at some Borel $A$, $\mathbb{P}^*_0 \in \mathcal{B}^\infty_\epsilon(\mathbb{P}_0)$, and $\mathbb{P}^*_1 \in \mathcal{B}^\infty_\epsilon(\mathbb{P}_1)$.*

See Frank & Niles-Weed (2024a, Theorem 1) for a proof of this statement. The maximizers $\mathbb{P}^*_0$, $\mathbb{P}^*_1$ can be interpreted as optimal adversarial attacks (see discussion following Frank & Niles-Weed (2024b, Theorem 7)). Frank (2024, Theorem 3.4) provide a criterion for uniqueness up to degeneracy in terms of dual maximizers.

**Theorem 6** (Frank (2025)). *The following are equivalent:*

*A) The adversarial Bayes classifier is unique up to degeneracy*

*B) There are maximizers $\mathbb{P}^*_0$, $\mathbb{P}^*_1$ of $\bar{R}$ for which $\mathbb{P}^*(\eta^* = 1/2) = 0$, where $\mathbb{P}^* = \mathbb{P}^*_0 + \mathbb{P}^*_1$ and $\eta^* = d\mathbb{P}^*_1/d\mathbb{P}^*$*

Thus, uniqueness corresponds to the situation in which the set where both classes are equally probable has measure zero under some optimal adversarial attack.

The analogous dual problem to $R^\epsilon_\phi$ uses $C^*_\phi(\eta)$ from (5)

$$\bar{R}_\phi(\mathbb{P}'_0, \mathbb{P}'_1) = \inf_{f \text{ Borel}} \int \phi(f) d\mathbb{P}'_1 + \int \phi(-f) d\mathbb{P}'_0 = \int C^*_\phi \left( \frac{d\mathbb{P}'_1}{d(\mathbb{P}'_1 + \mathbb{P}'_0)} \right) d(\mathbb{P}'_0 + \mathbb{P}'_1) \tag{13}$$

and the analogous minimax theorem states (Frank & Niles-Weed, 2024b, Theorem 6):

**Theorem 7** (Frank & Niles-Weed (2024b)). *Let $\bar{R}_\phi$ be defined as in (13). Then*

$$\inf_{\substack{f \text{ Borel,} \\ \mathbb{R}\text{-valued}}} R^\epsilon_\phi(f) = \sup_{\substack{\mathbb{P}'_1 \in \mathcal{B}^\infty_\epsilon(\mathbb{P}_1) \\ \mathbb{P}'_0 \in \mathcal{B}^\infty_\epsilon(\mathbb{P}_0)}} \bar{R}_\phi(\mathbb{P}'_0, \mathbb{P}'_1).$$

*with maximizers $\mathbb{P}^*_0 \in \mathcal{B}^\infty_\epsilon(\mathbb{P}_0)$, $\mathbb{P}^*_1 \in \mathcal{B}^\infty_\epsilon(\mathbb{P}_1)$ attained.*

Finally, optimal attacks for the surrogate problem are also optimal for the classification problem:

**Theorem 8.** *Consider maximizing the dual objectives $\bar{R}_\phi$ and $\bar{R}$ over $\mathcal{B}^\infty_\epsilon(\mathbb{P}_0) \times \mathcal{B}^\infty_\epsilon(\mathbb{P}_1)$.*

*1) If $\phi$ is consistent, then any maximizer $(\mathbb{P}^*_0, \mathbb{P}^*_1)$ of $\bar{R}_\phi$ over $\mathcal{B}^\infty_\epsilon(\mathbb{P}_0) \times \mathcal{B}^\infty_\epsilon(\mathbb{P}_1)$ also maximizes $\bar{R}$.*

*2) [Frank (2025)] If the adversarial Bayes classifier is unique up to degeneracy, then there exists a maximizer $(\mathbb{P}^*_0, \mathbb{P}^*_1)$ of $\bar{R}_\phi$ with $\mathbb{P}^*(\eta^* = 1/2) = 0$, where $\mathbb{P}^* = \mathbb{P}^*_0 + \mathbb{P}^*_1$ and $\eta^* = d\mathbb{P}^*_1/d\mathbb{P}^*$.*

See Appendix D for a proof of Item 1), Item 2) is shown in Theorems 5 and 7 of Frank (2025). This minimax machinery links the adversarial Bayes classifier, optimal attacks, and surrogate risks, establishing the dual formulations used in Section 3 to derive adversarial surrogate risk bounds.

## 3 Main Results

Prior work has characterized when a loss $\phi$ is adversarially consistent with respect to a distribution $\mathbb{P}_0$, $\mathbb{P}_1$. Theorem 3 shows that a distribution-independent surrogate risk bound is possible only when $C^*_\phi(1/2) < \phi(0)$. When $C^*_\phi(1/2) = \phi(0)$, Theorem 4 indicates that any such bound must depend on the marginal distribution of $\eta^*$ under $\mathbb{P}^*$, and moreover, is possible only if $\mathbb{P}^*(\eta^* = 1/2) = 0$.

Compare these statements with Proposition 1: Theorems 3, 4 and 8 together imply if either $C_\phi^*(1/2) < \phi(0)$ or if there exist some maximizers of $\bar{R}_\phi$ that satisfy Massart's noise condition, then $\phi$ is adversarially consistent for $\mathbb{P}_0$, $\mathbb{P}_1$. Alternatively, due to Theorem 8, one can equivalently assume that there are maximizers of $\bar{R}_\phi$ satisfying Massart's noise condition. Our first result extends Proposition 1 to the adversarial scenario, replacing $\mathbb{P}_0$, $\mathbb{P}_1$ with the distribution of optimal adversarial attacks.

**Theorem 9.** *Let $\phi$ be consistent and let $\mathbb{P}_0$, $\mathbb{P}_1$ be a distribution for which there are maximizers $\mathbb{P}_0^*$, $\mathbb{P}_1^*$ of the dual problem $\bar{R}_\phi$ that satisfy $|\eta^* - 1/2| \geq \alpha$ $\mathbb{P}^*$-a.e. for some constant $\alpha \in [0, 1/2]$ with $C_\phi^*(1/2 - \alpha) < \phi(0)$, where $\mathbb{P}^* = \mathbb{P}_0^* + \mathbb{P}_1^*$, $\eta^* = d\mathbb{P}_1^*/d\mathbb{P}^*$. Then*

$$R^\epsilon(f) - R_*^\epsilon \leq \frac{1}{\phi(0) - C_\phi^*(1/2 - \alpha)} \left( R_\phi^\epsilon(f) - R_{\phi,*}^\epsilon \right) \tag{14}$$

When $C_\phi^*(1/2) < \phi(0)$, setting $\alpha = 0$ in Theorem 9 yields a distribution-independent bound. As noted earlier, two losses satisfying this condition are the $\rho$-margin loss and the shifted sigmoid loss. Likewise, Theorem 1 ensures that the linear constant is finite whenever $\alpha \neq 0$ and $\phi$ is consistent.

The constant appearing in Theorem 9 is nearly optimal: Section 4.3 shows that it can be improved by at most a factor of two, and this gap is attained by a known counterexample to consistency. Thus, the result provides a sharp characterization of how tightly the adversarial classification risk can be controlled by the surrogate risk across all consistent convex losses.

Furthermore, the theorem parallels the classical realizable-case guarantee from the non-adversarial setting. If the optimal adversarial risk satisfies $R_*^\epsilon = 0$, then Massart's noise condition holds with $\alpha = 1/2$ (see Lemma 2). In this regime, Theorem 9 yields a linear relationship between adversarial classification and surrogate risks that is directly analogous to the non-adversarial bound in Proposition 1. Zero adversarial risk occurs whenever the supports of $\mathbb{P}_0$ and $\mathbb{P}_1$ are separated by at least $2\epsilon$ (Example 1 and Figure 3a).

Theorem 9 states that if some distribution of *optimal adversarial attacks* satisfies Massart's noise condition, then the excess adversarial surrogate risk is at worst a linear upper bound on the excess adversarial classification risk. However, if $C_\phi^*(1/2) = \phi(0)$, the bound's constant diverges as $\alpha \to 0$, reflecting the failure of adversarial consistency when the adversarial Bayes classifier is not unique up to degeneracy. For $\alpha \neq 1/2$, understanding the assumptions on $(\mathbb{P}_0, \mathbb{P}_1)$ which ensure Massart's condition for the distribution of adversarial attacks $(\mathbb{P}_0^*, \mathbb{P}_1^*)$ remains an open problem. Example 4.6 of Frank (2024) exhibits a distribution that satisfies Massart's noise condition and yet the adversarial Bayes classifier is not unique up to degeneracy. Thus Massart's noise condition for $\mathbb{P}_0, \mathbb{P}_1$ does not guarantee Massart's noise condition for $\mathbb{P}_0^*, \mathbb{P}_1^*$. See Example 2 and Figure 3b for an example where Theorem 9 applies with $\alpha > 0$.

One approach to relaxing the distributional restriction is to apply (14) only on the portion of the distribution where $|\eta^* - 1/2| \geq \alpha$ and then add back in the risk on $|\eta^* - 1/2| < \alpha$.

**Theorem 10.** *Assume that there exist maximizers $\mathbb{P}_0^*$, $\mathbb{P}_1^*$ of $\bar{R}_\phi$ that are induced by transport maps from $\mathbb{P}_0$, $\mathbb{P}_1$, and define $\mathbb{P}^* = \mathbb{P}_1^* + \mathbb{P}_0^*$, $\eta^* = d\mathbb{P}_1^*/d\mathbb{P}^*$. Let $0 \leq \alpha$, then*

$$R^\epsilon(f) - R_*^\epsilon \leq \frac{1}{\phi(0) - C_\phi^*(1/2 - \alpha)} \left( R_\phi^\epsilon(f) - R_{\phi,*}^\epsilon \right) + \left( \frac{1}{2} + \alpha \right) \mathbb{P}^*(|\eta^* - 1/2| < \alpha)$$

Since this holds for all $\alpha$, the right-hand side can be minimized over $\alpha$. Prior work from optimal transport theory verifies the assumption on $\mathbb{P}^*$ under mild conditions: Theorem 3.5 of Jylhä (2014) states that whenever $\mathbb{P}_0, \mathbb{P}_1$ are absolutely continuous with respect to Lebesgue measure and the norm $\|\cdot\|$ is strictly convex, the measures $\mathbb{P}_0^*, \mathbb{P}_1^*$ are induced by a transport map. It is unclear whether this holds for common datasets such as CIFAR-10 or MNIST.

Finally, an alternative approach to removing the distributional restriction is to average bounds of the form (14) over all values of $\eta^*$ yielding a distribution-dependent surrogate bound, valid whenever the adversarial Bayes classifier is unique up to degeneracy. For a given function $f$, let the *concave envelope* of $f$ be the smallest concave function larger than $f$:

$$\text{conc}(f) = \inf\{g :\geq f \text{ on } \text{dom}(f), g \text{ concave and upper semi-continuous}\} \tag{15}$$

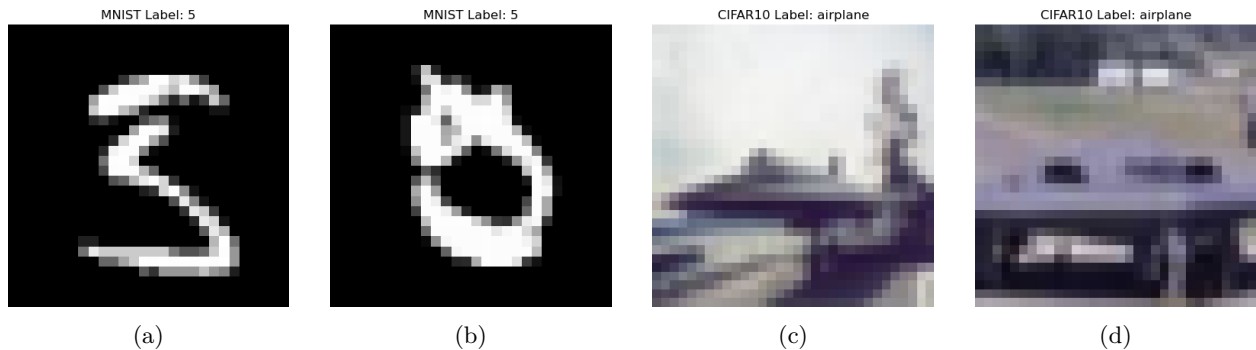

Figure 2: Ambiguous images in the MNIST and CIFAR10 datasets. (a) lies between a '5' and a '3', while (b) is difficult to classify at all, despite being labeled as a '5'. In CIFAR10, image (c) is ambiguous between a ship and an airplane, and image (d) is similarly hard to identify.

**Theorem 11.** *Assume $\mathbb{P}_0(\mathbb{R}^d) + \mathbb{P}_1(\mathbb{R}^d) \leq 1$, $\phi$ is a consistent loss with $C_\phi^*(1/2) = \phi(0)$, and the adversarial Bayes classifier is unique up to degeneracy. Let $\mathbb{P}_0^*$, $\mathbb{P}_1^*$ be maximizers of $\bar{R}_\phi$ for which $\mathbb{P}^*(\eta^* = 1/2) = 0$, with $\mathbb{P}^* = \mathbb{P}_0^* + \mathbb{P}_1^*$ and $\eta^* = d\mathbb{P}_1^*/d\mathbb{P}^*$. Define $H(z) = \text{conc}(\mathbb{P}^*(|\eta^* - 1/2| \leq z))$, $\Psi$ as Theorem 2, and let $\tilde{\Lambda}(z) = \Psi^{-1}(\min(\frac{z}{4}, \phi(0)))$. Then*

$$R^\epsilon(f) - R_*^\epsilon \leq \tilde{\Phi}(R_\phi^\epsilon(f) - R_{\phi,*}^\epsilon)$$

*with*

$$\tilde{\Phi}(z) = 4\left(\text{id} + \min(1, \sqrt{-eH \ln H})\right) \circ \tilde{\Lambda}$$

This theorem is established under the assumption $\mathbb{P}_0(\mathbb{R}^d) + \mathbb{P}_1(\mathbb{R}^d) \leq 1$, which serves as an essential intermediate step for extending the result to case where the adversarial Bayes classifier is not uniquely defined up to degeneracy. See Example 3 and Figure 3c for an example of calculating a distribution-dependent surrogate risk bound.

The function $H$ is always continuous and satisfies $H(0) = 0$, ensuring that this bound is non-vacuous (see Lemma 7 in Section 5). Further notice that $H \ln H$ approaches zero as $H \to 0$.

The map $\tilde{\Phi}$ combines two components: $\tilde{\Lambda}$, a modified version of $\Psi^{-1}$, and $H$, a modification of the cdf of $|\eta^* - 1/2|$. The function $\tilde{\Lambda}$ is a scaled version of $\Psi^{-1}$, where $\Psi$ is the surrogate risk bound in the non-adversarial case of Theorem 2. The domain of $\Psi^{-1}$ is $[0, \phi(0)]$, and thus the role of the min in the definition of $\tilde{\Lambda}$ is to truncate the argument so that it fits into this domain. The factor of $1/4$ in this function appears to be an artifact of our proof, see Section 5 for further discussion. In contrast, the map $H$ translates the distribution of $\eta^*$ into a surrogate risk transformation. Compare with Theorem 4, which states that consistency fails if $\mathbb{P}^*(\eta^* = 1/2) > 0$; accordingly, the function $H$ becomes a poorer bound when more mass of $\eta^*$ is near $1/2$.

If $\mathbb{P}^*(\eta^* = 1/2)$ is small, this result can still provide an informative surrogate bound.

**Theorem 12.** *Assume that there exist maximizers $\mathbb{P}_0^*$, $\mathbb{P}_1^*$ of $\bar{R}_\phi$ that are induced by transport maps from $\mathbb{P}_0^*$, $\mathbb{P}_1^*$, and define $\mathbb{P}^* = \mathbb{P}_1^* + \mathbb{P}_0^*$, $\eta^* = d\mathbb{P}_1^*/d\mathbb{P}^*$. Let $\tilde{\Phi}$ be the function in Theorem 11, but with $H$ defined as $H(z) = \text{conc}(\mathbb{P}^*(0 < |\eta^* - 1/2| \leq z))$. Then*

$$R^\epsilon(f) - R_*^\epsilon \leq \tilde{\Phi}(R_\phi^\epsilon(f) - R_{\phi,*}^\epsilon) + \frac{\mathbb{P}^*(\eta^* = 1/2)}{2}$$

Removing the assumption that $\mathbb{P}_0^*, \mathbb{P}_1^*$ are induced by a transport map from Theorems 10 and 12 remains an open problem. We conjecture that this assumption is, in fact, unnecessary.

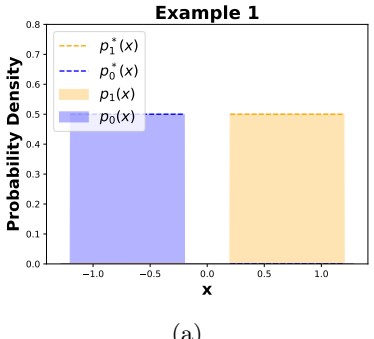 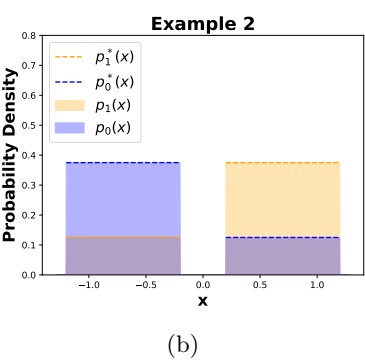 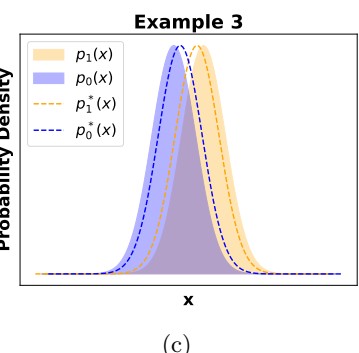

Figure 3: Distributions from Examples 1 to 3 along with attacks that maximize the dual $\bar{R}_\phi$.

**Comparison with real-world datasets**

Experimental results from prior work suggest that, in real-world datasets, $\eta^*$ is typically concentrated near 0 and 1. Bhagoji et al. (2019) compute lower bounds on the adversarial classification risk for binary tasks, focusing on classifying digits '3' and '7' in MNIST under $\ell_2$ perturbations. Their lower bound remains close to 0 for $\epsilon \leq 3$ and increases to 0.2 at $\epsilon = 4$. Since $C^*(\eta^*)$ attains its maximum at $\eta^* = 1/2$, a small adversarial risk implies that the distribution places little mass in a neighborhood of $|\eta^* - 1/2| = 0$. Similar trends are observed on Fashion MNIST and CIFAR10. Dai et al. (2023) extend these bounds to the multiclass setting, though extending adversarial surrogate bounds beyond binary classification remains an open problem.

When the optimal adversarial risk is non-zero, the adversarial Bayes classifier may not be unique up to degeneracy. Even without adversarial perturbations, datasets like MNIST and CIFAR10 contain inherently ambiguous examples. Northcutt et al. (2021) identify such cases, four are depicted in Figure 2. One would expect $\eta(\mathbf{x}) = 1/2$ for such examples. Bartoldson et al. (2024) show that similar ambiguity arises in adversarial settings: under $\ell_\infty$ perturbations of size $8/255$, approximately 6% of adversarial examples are ambiguous in the CIFAR10 dataset. In the binary scenario, one would thus expect $\eta^*(x) = 1/2$ for these inputs, and thus one must apply Theorem 10 or Theorem 12. Extending the concept of uniqueness of the adversarial Bayes classifier to multiclass settings remains an open problem.

**Examples**

Below we present three examples illustrating the applicability of our main theorems. All examples involve one-dimensional distributions, and we denote the pdfs of $\mathbb{P}_0$ and $\mathbb{P}_1$ by $p_0$ and $p_1$.

To start, if $R_*^\epsilon = 0$ then $\eta^* \in \{0, 1\}$ $\mathbb{P}^*$-a.e. for any maximizers of $\bar{R}_\phi$. Therefore, for any such distribution, the optimal attack satisfies Massart's noise condition with $\alpha = 1/2$, see Appendix J.1 for a proof.

**Lemma 2.** *Assume* $R_*^\epsilon = 0$, *let* $(\mathbb{P}_0^*, \mathbb{P}_1^*)$ *maximize* $\bar{R}_\phi$, *and define* $\mathbb{P}^* = \mathbb{P}_0^* + \mathbb{P}_1^*$, $\eta^* = d\mathbb{P}_1^*/d\mathbb{P}^*$. *Then* $\mathbb{P}^*(\eta^* \in \{0, 1\}) = 1$.

Any distribution for which the supports of $\mathbb{P}_0$, $\mathbb{P}_1$ are more than $2\epsilon$ apart must have zero risk.

**Example 1** (When $R_*^\epsilon = 0$)**.** Let

$$p_0(x) = \begin{cases} 1 & \text{if } x \in [-1-\delta, -\delta] \\ 0 & \text{otherwise} \end{cases} \qquad p_1(x) = \begin{cases} 1 & \text{if } x \in [\delta, 1+\delta] \\ 0 & \text{otherwise} \end{cases}$$

for some $\delta > 0$. See Figure 3a for a depiction. This distribution satisfies $R_*^\epsilon = 0$ for all $\epsilon \leq \delta$ and thus Lemma 2 implies that the surrogate bound of Theorem 9 applies.

Examples 2 and 3 require computing maximizers to the dual $\bar{R}_\phi$; See Appendices J.2 and J.3 for these calculations. The following example illustrates a distribution for which Massart's noise condition can be verified for a distribution of optimal attacks.

**Example 2** (Massart's noise condition). Let $\delta > 0$ and let $p$ be the uniform density on $[-1-\delta, -\delta] \cup [\delta, 1+\delta]$. Define $\eta$ by

$$\eta(x) = \begin{cases} \frac{1}{4} & \text{if } x \in [-1-\delta, -\delta] \\ \frac{3}{4} & \text{if } x \in [\delta, 1+\delta] \end{cases} \tag{16}$$

see Figure 3b for a depiction of $p_0$ and $p_1$. For this distribution and $\epsilon \leq \delta$, the minimal surrogate and adversarial surrogate risks are always equal ($R_{\phi,*} = R_{\phi,*}^\epsilon$). This fact together with Theorem 7 imply that optimal attacks on this distribution are $\mathbb{P}_1^* = \mathbb{P}_1$ and $\mathbb{P}_0^* = \mathbb{P}_0$, see Appendix J.2 for details. Consequently: the distribution of optimal attacks $\mathbb{P}_0^*$, $\mathbb{P}_1^*$ satisfies Massart's noise condition with $\alpha = 1/4$ and as a result the bounds of Theorem 9 apply. When $\epsilon \in (\delta, 1 + \delta)$, pdfs of the distributions that maximize the dual are $p_1^*(x) = p_1(x+\epsilon)$, $p_0^*(x) = p_0(x-\epsilon)$, where $p_1(x) = \eta(x)p(x)$ and $p_0(x) = (1-\eta(x))p(x)$. These distributions satisfy $\mathbb{P}^*(\eta = 1/2) = (\epsilon - \delta)$ while $\mathbb{P}^*(|\eta - 1/2| \geq 1/4) = 1 - (\epsilon - \delta)$. Thus Theorem 10 provides a surrogate bound.

The final example presents a case in which Massart's noise condition fails for the distribution of optimal adversarial attacks, yet the adversarial Bayes classifier remains unique up to degeneracy. Theorem 11 still yields an informative surrogate bound.

**Example 3** (Gaussian example). Consider an equal-variance Gaussian mixture with $\mu_0 + 2\epsilon < \mu_1 < \mu_0 + \sqrt{2}\sigma$:

$$p_0(x) = \frac{1}{2} \cdot \frac{1}{\sqrt{2\pi}\sigma} e^{-\frac{(x-\mu_0)^2}{2\sigma^2}}, \quad p_1(x) = \frac{1}{2} \cdot \frac{1}{\sqrt{2\pi}\sigma} e^{-\frac{(x-\mu_1)^2}{2\sigma^2}},$$

see Figure 3c for a depiction. The optimal attacks $\mathbb{P}_0^*$, $\mathbb{P}_1^*$ are gaussians centered at $\mu_0 + \epsilon$ and $\mu_1 - \epsilon$ respecively, with pdfs

$$p_0^*(x) = \frac{1}{2} \cdot \frac{1}{\sqrt{2\pi}\sigma} e^{-\frac{(x-(\mu_0+\epsilon))^2}{2\sigma^2}}, \quad p_1^*(x) = \frac{1}{2} \cdot \frac{1}{\sqrt{2\pi}\sigma} e^{-\frac{(x-(\mu_1-\epsilon))^2}{2\sigma^2}}. \tag{17}$$

We verify that $\mathbb{P}_0^*$ and $\mathbb{P}_1^*$ are in fact optimal by finding a function $f^*$ for which $R_\phi^\epsilon(f^*) = \bar{R}_\phi(\mathbb{P}_0^*, \mathbb{P}_1^*)$, the strong duality result in Theorem 7 will then imply that $\mathbb{P}_0^*$ and $\mathbb{P}_1^*$ must maximize the dual $\bar{R}_\phi$, see Appendix J.3 for details.

Further, when $\mu_1 - \mu_0 \leq \sqrt{2}\sigma$, then the function $h(z) = \mathbb{P}^*(|\eta^* - 1/2| \leq z)$ is concave in $z$ and consequently $H = h$, see Appendix J.4 for details. Although $h$ is unwieldy function, comparison to its linear approximation at zero gives the bound

$$H(z) \leq \min\left(\frac{16\sigma^2}{\mu_1 - \mu_0 - 2\epsilon} z, 1\right). \tag{18}$$

Again, see Appendix J.4 for details.

When $\epsilon \geq (\mu_1 - \mu_0)/2$, Frank (2024, Example 4.1) demonstrates that the adversarial Bayes classifier is not unique up to degeneracy. Notably, the bound in preceding example deteriorates as $(\mu_1 - \mu_0)/2 \to \epsilon$, and then fails entirely when $\epsilon = (\mu_1 - \mu_0)/2$.

## 4 Proof of Linear Surrogate Bounds

### 4.1 Proof of Theorem 9

The proof of Theorem 9 relies on decomposing the excess adversarial classification and surrogate risks into non-negative terms, revealing their structural similarity and allowing for a pointwise comparison.

Let $\mathbb{P}_0^*, \mathbb{P}_1^*$ be any maximizers of $\bar{R}_\phi$. These distributions also maximize $\bar{R}$ by Theorem 8. Set $\mathbb{P}^* = \mathbb{P}_0^* + \mathbb{P}_1^*$, $\eta^* = d\mathbb{P}_1^*/d\mathbb{P}^*$. Let $\gamma_0^*, \gamma_1^*$ be couplings between $\mathbb{P}_0$, $\mathbb{P}_0^*$ and $\mathbb{P}_1$, $\mathbb{P}_1^*$ achieving the $W_\infty$ distances (11). The excess classification risk can be decomposed as

$$R^\epsilon(f) - R_*^\epsilon = R^\epsilon(f) - \bar{R}(\mathbb{P}_0^*, \mathbb{P}_1^*) = \int i_1(f) d\gamma_1^* + \int i_0(f) d\gamma_0^* \tag{19}$$

with

$$i_1(f) = \big(S_\epsilon(\mathbf{1}_{f\leq 0})(\mathbf{x}) - \mathbf{1}_{f\leq 0}(\mathbf{x}')\big) + \big(C(\eta^*, f) - C^*(\eta^*)\big)$$
$$i_0(f) = \big(S_\epsilon(\mathbf{1}_{f>0})(\mathbf{x}) - \mathbf{1}_{f>0}(\mathbf{x}')\big) + \big(C(\eta^*, f) - C^*(\eta^*)\big).$$

The first term measures the discrepancy between the worst-case attack on $f$ and the attack induced by $\mathbb{P}_0^*, \mathbb{P}_1^*$, the optimal attack for the distribution $\mathbb{P}_0, \mathbb{P}_1$. The second term measures the excess conditional risk under the optimal attack $\mathbb{P}_0^*, \mathbb{P}_1^*$. Lemma 1 implies that $S_\epsilon(\mathbf{1}_{f\leq 0})(\mathbf{x}) - \mathbf{1}_{f\leq 0}(\mathbf{x}')$ must be positive, while the definition of $C^*$ implies that $C(\eta^*, f) - C^*(\eta^*) \geq 0$.

Similarly, one can express the excess surrogate risk as

$$R_\phi^\epsilon(f) - R_{\phi,*}^\epsilon = \int i_1^\phi(f) d\gamma_1^* + \int i_0^\phi(f) d\gamma_0^* \tag{20}$$

with

$$i_1^\phi(f) = \big(S_\epsilon(\phi(f))(\mathbf{x}) - \phi(f)(\mathbf{x}')\big) + \big(C_\phi(\eta^*, f) - C_\phi^*(\eta^*)\big)$$
$$i_0^\phi(f) = \big(S_\epsilon(\phi(-f))(\mathbf{x}) - \phi(-f)(\mathbf{x}')\big) + \big(C_\phi(\eta^*, f) - C_\phi^*(\eta^*)\big)$$

The following lemma is the core inequality linking $i_k$ to $i_k^\phi$ under Massart's noise condition. It shows that each classification-risk term can be bounded by a constant multiple of its surrogate risk counterpart.

**Lemma 3.** *Define $i_0^\phi, i_1^\phi$ as in* (20) *and assume that the distribution of optimal adversarial attacks $\mathbb{P}_0^*$, $\mathbb{P}_1^*$ satisfies Massart's noise condition. Then*

$$i_0(f) \leq \frac{1}{\phi(0) - C_\phi^*(1/2 - \alpha)} i_0^\phi(f). \tag{21} \qquad i_1(f) \leq \frac{1}{\phi(0) - C_\phi^*(1/2 - \alpha)} i_1^\phi(f). \tag{22}$$

*hold $\gamma_0^*$-a.e. and $\gamma_1^*$-a.e. respectively.*

Lemma 3 directly implies Theorem 9 by integration over couplings $\gamma_1^*, \gamma_0^*$.

*Proof of Theorem 9.* Combine (19), Lemma 3, and (20). $\qquad\square$

## 4.2 Proof of Lemma 3

The proof proceeds by partitioning the domain $\mathbb{R}^d \times \mathbb{R}^d$ into regions where the supremum-based classification either matches $(D_k)$ or exceeds $(E_k)$ the decision under the optimal attack. On each region, we derive a separate bound relating $i_k$ and $i_k^\phi$. Define the sets $D_k$, $E_k$,

$$D_0 = \{(\mathbf{x}, \mathbf{x}') : S_\epsilon(\mathbf{1}_{f>0})(\mathbf{x}) - \mathbf{1}_{f(\mathbf{x}')} = 0\} \tag{23}$$
$$E_0 = \{(\mathbf{x}, \mathbf{x}') : S_\epsilon(\mathbf{1}_{f>0})(\mathbf{x}) - \mathbf{1}_{f(\mathbf{x}')>0} = 1\} \tag{24}$$
$$D_1 = \{(\mathbf{x}, \mathbf{x}') : S_\epsilon(\mathbf{1}_{f\leq 0})(\mathbf{x}) - \mathbf{1}_{f(\mathbf{x}')\leq 0} = 0\} \tag{25}$$
$$E_1 = \{(\mathbf{x}, \mathbf{x}') : S_\epsilon(\mathbf{1}_{f\leq 0})(\mathbf{x}) - \mathbf{1}_{f(\mathbf{x}')\leq 0} = 1\} \tag{26}$$

By construction, $D_1 \cup E_1 = \mathbb{R}^d \times \mathbb{R}^d$ and $D_0 \cup E_0 = \mathbb{R}^d \times \mathbb{R}^d$.

The following lemma records a simple but useful structural property of $E_0$ and $E_1$, which allows us to bound the surrogate loss terms from below on these sets.

**Lemma 4.** *Let $E_k$ be as in Equations* (24) *and* (26)*. Then $S_\epsilon(\mathbf{1}_{f>0})(\mathbf{x}) = \mathbf{1}_{f>0}(\mathbf{x}') = 1$ $\gamma_1^*$-a.e. on $E_1$ while $S_\epsilon(\mathbf{1}_{f\leq 0})(\mathbf{x}) = \mathbf{1}_{f\leq 0}(\mathbf{x}') = 1$ $\gamma_0^*$-a.e. on $E_0$.*

*Proof.* We'll prove the statement for $E_1$, the argument for $E_0$ is analogous. Specifically, we will show that one cannot simultaneously have $S_\epsilon(\mathbf{1}_{f\leq 0})(\mathbf{x}) - \mathbf{1}_{f\leq 0}(\mathbf{x}') = 1$ and $S_\epsilon(\mathbf{1}_{f>0})(\mathbf{x}) - \mathbf{1}_{f>0}(\mathbf{x}') = 1$.

Consider $(\mathbf{x}, \mathbf{x}') \in E_1$: as both $S_\epsilon(\mathbf{1}_{f\leq 0})(\mathbf{x})$ and $\mathbf{1}_{f\leq 0}(\mathbf{x}')$ are 0-1 valued, the relation $S_\epsilon(\mathbf{1}_{f\leq 0})(\mathbf{x}) - \mathbf{1}_{f(\mathbf{x}')\leq 0} = 1$ implies that $\mathbf{1}_{f(\mathbf{x}')\leq 0} = 0$ and thus $\mathbf{1}_{f(\mathbf{x}')>0} = 1$. The fact that $S_\epsilon(\mathbf{1}_{f>0})(\mathbf{x}) \geq \mathbf{1}_{f>0}(\mathbf{x}')$ on supp $\gamma_1^*$ and supp $\gamma_1^* \subset \Delta_\epsilon$ implies that $S_\epsilon(\mathbf{1}_{f>0})(\mathbf{x}) = 1$ $\gamma_1^*$-a.e. on $E_1$.

$\qquad\square$

The next result bounds the terms $i_k^\phi$ from below.

**Lemma 5.** *The relations (27) and (28) hold on $E_0$ and $E_1$ respectively.*

$$i_0^\phi(f) \geq \phi(0) - C_\phi^*(\eta^*). \qquad (27) \qquad\qquad i_1^\phi(f) \geq \phi(0) - C_\phi^*(\eta^*). \qquad (28)$$

*Proof.* We will prove the statement for $E_1$, the argument for $E_0$ is analogous. Observe that

$$i_1^\phi(f) = S_\epsilon(\phi(f))(\mathbf{x}) + (1 - \eta^*)(\phi(-f(\mathbf{x}')) - \phi(f(\mathbf{x}'))) - C_\phi^*(\eta^*)$$

Now as $S_\epsilon(\mathbf{1}_{f \leq 0})(\mathbf{x}) = 1$, one can conclude that there is a point in $\mathbf{z} \in \overline{B_\epsilon(\mathbf{x})}$ for which $f(\mathbf{z}) \leq 0$, and thus $S_\epsilon(\phi(f))(\mathbf{x}) \geq \phi(0)$. Next, Lemma 4 implies that $f(\mathbf{x}') > 0$ and hence $\phi(-f(\mathbf{x}')) - \phi(f(\mathbf{x}')) \geq 0$. Therefore, one can conclude (28).

$\square$

Furthermore, a simple calculation bounds the $i_k$ from above.

**Lemma 6.** *On the set $D_k$*

$$i_k(f) = C(\eta^*, f) - C^*(\eta^*) \qquad (29)$$

*while on $E_k$*

$$i_k(f) = 1 + C(\eta^*, f) - C^*(\eta^*) \qquad (30)$$

*Proof.* We will show the statement $k = 1$ the argument for $k = 0$ is analogous. On $D_1$, $S_\epsilon(\mathbf{1}_{f \leq 0})(\mathbf{x}) - \mathbf{1}_{f(\mathbf{x}') \leq 0} = 0$, implying (29). Similarly, on $E_1$, $S_\epsilon(\mathbf{1}_{f \leq 0})(\mathbf{x}) - \mathbf{1}_{f(\mathbf{x}') \leq 0} = 1$, implying (30).

$\square$

Comparing the upper and lower bounds present in Lemmas 4 and 6 proves Lemma 3.

*Proof of Lemma 3.* We will discuss (22), the argument for (21) is analogous. We prove the bound separately on $D_1$ and $E_1$, whose union is $\mathbb{R}^d$. First, notice that (8) implies that

$$C(\eta^*(\mathbf{x}'), f(\mathbf{x}')) - C^*(\eta^*(\mathbf{x}')) \leq \frac{1}{\phi(0) - C_\phi^*(1/2 - \alpha)} \left( C_\phi(\eta^*(\mathbf{x}'), f(\mathbf{x}')) - C_\phi^*(\eta^*(\mathbf{x}')) \right) \quad \mathbb{P}^*\text{-a.e.} \qquad (31)$$

**On the set $D_1$:**
Lemma 6 implies that

$$i_1(f) = C(\eta^*(\mathbf{x}'), f(\mathbf{x}')) - C^*(\eta^*(\mathbf{x}'))$$

and thus the desired inequality follows from (31) and the fact that $S_\epsilon(\phi \circ f)(\mathbf{x}) - \phi \circ f(\mathbf{x}') \geq 0$ $\gamma_1^*$-a.e.

**On the set $E_1$:**
On $E_1$,

$$i_1(f) = 1 + C(\eta^*(\mathbf{x}'), f(\mathbf{x}')) - C^*(\eta^*(\mathbf{x}'))$$

However, due to Lemma 5,

$$S_\epsilon(\mathbf{1}_{f \leq 0})(\mathbf{x}) - \mathbf{1}_{f \leq 0}(\mathbf{x}') = 1 = \frac{\phi(0) - C_\phi^*(\eta^*)}{\phi(0) - C_\phi^*(\eta^*)} \leq \frac{1}{\phi(0) - C_\phi^*(\eta^*)} (S_\epsilon(\phi \circ f)(\mathbf{x}) - \phi(f(\mathbf{x}'))) \qquad (32)$$

The last inequality is a consequence of the assumption $|\eta^* - 1/2| \leq \alpha$. Summing this relation with (31) shows (22). $\square$

### 4.3 Lower Bounds

The bound in Theorem 9 provides a general guarantee relating the adversarial classification and surrogate risks. We now show that this bound cannot be substantially improved.

This example describes functions in which the worst-case attack and the attack induced by $\mathbb{P}_0^*$, $\mathbb{P}_1^*$ differ substantially. This function sequence is the counterexample to consistency proposed in prior work (Meunier et al., 2022; Li & Telgarsky, 2023). Intuitively, the sign flip causes the classifier to misclassify both classes, even though a constant function would achieve lower risk for this distribution.

**Example 4** (Lower bound for Theorem 9). Let $\phi$ satisfy $C_\phi^*(1/2) = \phi(0)$, and consider a distribution supported on $[-\epsilon, \epsilon]$ with $\mathbb{P}^*(\eta = 1/2 + \alpha) = 1$. Define the sequence of functions

$$f_n = \begin{cases} \frac{1}{n} & \mathbf{x} \neq 0 \\ -\frac{1}{n} & \mathbf{x} = 0 \end{cases} \tag{33}$$

For this sequence, $R^\epsilon(f_n) - R_*^\epsilon = 1/2 + \alpha$ while the adversarial surrogate risk converges to $\lim_{n\to\infty} R_\phi^\epsilon(f_n) = \phi(0) - C_\phi^*(1/2 + \alpha)$. Consequently,

$$\lim_{n\to\infty} \frac{R^\epsilon(f_n) - R_*^\epsilon}{R_\phi^\epsilon(f_n) - R_{\phi,*}^\epsilon} = \frac{\frac{1}{2} + \alpha}{\phi(0) - C_\phi^*(1/2 - \alpha)}.$$

It follows that for any $\delta > 0$ there exists $f$ such that

$$R^\epsilon(f) - R_*^\epsilon \geq \frac{1/2 + \alpha}{\phi(0) - C_\phi^*(1/2 - \alpha)} \left( R_\phi^\epsilon(f) - R_{\phi,*}^\epsilon \right) - \delta.$$

In particular, the constant in Theorem 9 is overestimated by factor of at most $1/(1/2 + \alpha) \leq 2$. However, this example demonstrates that Theorem 9 is tight when $\alpha = 1/2$.

The constant in Theorem 9 is known to be sub-optimal when $\alpha < 1/2$. In particular, Theorem 4 of Frank (2025) proves that $R^\epsilon(f) - R_*^\epsilon \leq (1/2)/(\phi(0) - C_{\phi_\rho}^*(1/2))(R_{\phi_\rho}^\epsilon(f) - R_{\phi_\rho,*}^\epsilon)$ for the $\rho$-margin loss $\phi_\rho(\alpha) = \min(1, \max(0, 1 - \alpha/\rho))$. We conjecture that the tight constant in Theorem 9 is in fact $(1/2 + \alpha)/(\phi(0) - C_\phi^*(1/2 - \alpha))$. Together, these observations indicate that the bound in Theorem 9 captures the correct order of dependence on $\alpha$ and $\phi$, and that only the numerican constant can potentially be improved.

## 5 Proof of Theorem 11

Before proving Theorem 11, we will show that this bound is non-vacuous when the adversarial Bayes classifier is unique up to degeneracy. The function $h(z) = \mathbb{P}(|\eta^* - 1/2| \leq z)$ is a cdf, and is thus right-continuous in $z$. Furthermore, if the adversarial Bayes classifier is unique up to degeneracy, then $h(0) = 0$. The following lemma implies that if $H = \text{conc}(h)$ then $H$ is continuous at 0 with $H(0) = 0$. See Appendix E for a proof. This result implies that the bound in Theorem 11 is non-vacuous.

**Lemma 7.** Let $h : [0, 1/2] \to \mathbb{R}$ be a non-decreasing function with $h(0) = 0$ and $h(1/2) = 1$ that is right-continuous at 0. Then $\text{conc}(h)$ is non-decreasing, continuous on $[0, 1/2]$, and $\text{conc}(h)(0) = 0$.

The first step in proving Theorem 11 is showing an analog of Theorem 9 with $\alpha = 0$ for which the linear function is replaced by an $\eta$-dependent concave function.

**Proposition 2.** Let $\Phi$ be a concave non-decreasing function for which $C(\eta, \alpha) - C^*(\eta) \leq \Phi(C_\phi(\eta, \alpha) - C_\phi^*(\eta))$ for any $\eta \in [0, 1]$ and $\alpha \in \overline{\mathbb{R}}$. Let $\mathbb{P}_0^*$, $\mathbb{P}_1^*$ be any two maximzers of $\bar{R}_\phi$ for which $\mathbb{P}^*(\eta^* = 1/2) = 0$, where $\mathbb{P}^* = \mathbb{P}_0^* + \mathbb{P}_1^*$ and $\eta^* = d\mathbb{P}_1^*/d\mathbb{P}^*$. Let $G : [0, \infty) \to \mathbb{R}$ be any non-decreasing concave function for which the quantity

$$K = \int \frac{1}{G(\phi(0) - C_\phi^*(\eta^*))} d\mathbb{P}^*$$

*is finite. Then $R^\epsilon(f) - R^\epsilon_* \leq \tilde{\Phi}(R^\epsilon_\phi(f) - R^\epsilon_{\phi,*})$, where*

$$\tilde{\Phi}(z) = 4\sqrt{KG\left(\frac{1}{4}z\right)} + 2\Phi\left(\frac{1}{2}z\right) \tag{34}$$

The proof strategy mirrors that of Theorem 9, but with $\Phi$ and $G$ replacing the fixed constant bound.

Uniqueness up to degeneracy and Theorem 1 guarantee that the denominator $\phi(0) - C^*_\phi(\eta^*)$ is strictly positive $\mathbb{P}^*$-a.e. The function $\Psi^{-1}$ in Theorem 2 and the surrogate bounds of Zhang (2004) provide examples of candidate functions for $\Phi$. As before, this result is proved by dividing the risks $R^\epsilon_\phi$, $R^\epsilon$ as the sum of four terms as in (19), (20) and then bounding these quantities over the sets $D_k$, $E_k$ defined in (25),(26) separately.

The factor of 1/4 in (34) arises as an artifact of the proof technique: one factor of 2 from averaging over two sets $D_1$, $E_1$, (see (56) in Appendix F), and another factor of 2 from combining the bounds associated with the two integrals corresponding to class 0 and class 1(see Equations (54) and (56) in Appendix F).

We now turn to the problem of identifying functions $G$ for which the constant $K$ in the preceding proposition is guaranteed to be finite when the adversarial Bayes classifier is unique, but distribution dependent. Observe that if $h$ is the cdf of $|\eta - 1/2|$ and $h$ is continuous, then $\int 1/h^r dh$ is always finite for $r \in (0,1)$. This calculation suggests $\Phi = h \circ \Psi^{-1}$, with $\Psi$ defined in Theorem 2. To ensure the concavity of $G$, we instead select $G = H \circ \Psi^{-1}$ with $H = \mathrm{conc}(h)$.

**Lemma 8.** *Assume $C^*_\phi(1/2) = \phi(0)$. Let $\mathbb{P}_1$, $\mathbb{P}_0$, $\mathbb{P}^*_1$, $\mathbb{P}^*_0$, $\phi$, $H$, and $\Psi$ be as in Theorem 11. Let $\Lambda(z) = \Psi^{-1}(\min(z, \phi(0)))$. Then for any $r \in (0,1)$,*

$$R^\epsilon(f) - R^\epsilon_* \leq \tilde{\Phi}(R^\epsilon_\phi(f) - R^\epsilon_{\phi,*}) \tag{35}$$

*with*

$$\tilde{\Phi}(z) = 4\sqrt{\frac{1}{1-r}H\left(\frac{1}{2}\Lambda\left(\frac{1}{4}z\right)\right)^r} + 2\Lambda\left(\frac{z}{2}\right). \tag{36}$$

*Proof.* For convenience, let $G = (H \circ \frac{1}{2}\Lambda)^r$. Then $G$ is concave because it is the composition of a concave function and an increasing concave function. We will verify that $K$ is finite and yields the constant in the bound:

$$K = \int \frac{1}{G(\phi(0) - C^*_\phi(\eta^*))} d\mathbb{P}^* \leq \frac{1}{1-r}$$

First,

$$\int \frac{1}{G(\phi(0) - C^*_\phi(\eta^*))} d\mathbb{P}^* = \int \frac{1}{H(|\eta^* - 1/2|)^r} d\mathbb{P}^* = \int_{[0,\frac{1}{2}]} \frac{1}{H(s)^r} d\mathbb{P}^* \sharp s = \int_{(0,\frac{1}{2}]} \frac{1}{H(s)^r} d\mathbb{P}^* \sharp s$$

with $s = |\eta^* - 1/2|$. The assumption $\mathbb{P}^*(|\eta^* = 1/2|) = 0$ allows us to drop 0 from the domain of integration. Because the function $H$ is continuous on $(0,1]$ by Lemma 7, this last expression can be evaluated as a Riemann-Stieltjes integral with respect to the function $h(s) = \mathbb{P}(|\eta^* - 1/2| \leq s)$:

$$\int_{(0,\frac{1}{2}]} \frac{1}{H(s)^r} d\mathbb{P}^* \sharp s = \int_0^{1/2} \frac{1}{H(s)^r} dh \tag{37}$$

This result is standard when $\mathbb{P}^*$ is Lebesgue measure, (see for instance Theorem 5.46 of Wheeden & Zygmund (1977)). We prove equality in (37) for strictly decreasing functions in Proposition 4 in Appendix G.1.

Finally, the integral in (37) can be bounded as

$$\int \frac{1}{H(s)^r} dh \leq \frac{1}{1-r} \tag{38}$$

If $h$ were differentiable, then the chain rule would imply

$$\int \frac{1}{H(s)^r} dh \leq \int \frac{1}{h(s)^r} dh = \int_0^{\frac{1}{2}} \frac{1}{h(s)^r} h'(s) dz = \frac{1}{1-r} h(s)^{1-r} \Big|_0^{\frac{1}{2}} \leq \frac{1}{1-r}.$$

This calculation is more delicate for non-differentiable $H$; we formally prove inequality in (38) in Appendix G.2.

This calculation proves the inequality (35) with $\tilde{\Phi}$ as (36) $\qquad\qquad \square$

To obtain the bound in Theorem 11, first observe that the concavity of $\Lambda$ together with the fact that $\Lambda(0) = 0$ implies that $2\Lambda(z/2) \leq 4\Lambda(z/4)$. Next, minimizing the bound in Lemma 8 over $r$ then produces Theorem 11, see Appendix H for details.

## 6 Proof of Theorems 10 and 12

The main insight behind Theorems 10 and 12 is that a transport map that realizes the optimal adversarial perturbations also preserves optimality when restricted to certain subsets of $\mathbb{R}^d$, allowing a reduction from the global to a local problem in both the dual and primal formulations. The following lemma formalizes the fact that under a transport map structure, restricting the primal problem to the pre-image of a set $Q$ corresponds exactly to restricting the dual maximizers to $Q$ itself.

**Lemma 9.** *Let $\mathbb{P}_0, \mathbb{P}_1$ be a data distribution and let $\mathbb{P}_0^* \in \mathcal{B}_\epsilon^\infty(\mathbb{P}_0), \mathbb{P}_1^* \in \mathcal{B}_\epsilon^\infty(\mathbb{P}_1)$ maximize $\bar{R}_\phi$. Assume there exists transport maps $T_0, T_1$ for which $\mathbb{P}_i^* = \mathbb{P}_i \sharp T_i$ with $\|T_i(\mathbf{x}) - \mathbf{x}\| \leq \epsilon$. Let $Q$ be any set and define $U_i = T_i^{-1}(Q)$.*

*If the data is distributed according to $\mathbb{P}_0|_{U_0}, \mathbb{P}_1|_{U_1}$, then $\mathbb{P}_0^*|_Q, \mathbb{P}_1^*|_Q$ maximize $\bar{R}_\phi$ over $\mathcal{B}_\epsilon^\infty(\mathbb{P}_0|_{U_0}) \times \mathcal{B}_\epsilon^\infty(\mathbb{P}_1|_{U_1})$.*

In the remainder of this section it will be useful to include the data distribution in the notation for the primal problem. Thus, for the remainder of this section, we define

$$R_\phi^\epsilon(f; \mathbb{P}_0, \mathbb{P}_1) = \int S_\epsilon(\phi \circ f) d\mathbb{P}_1 + \int S_\epsilon(\phi \circ -f) d\mathbb{P}_0 \quad R^\epsilon(f; \mathbb{P}_0, \mathbb{P}_1) = \int S_\epsilon(\mathbf{1}_{f \leq 0}) d\mathbb{P}_1 + \int S_\epsilon(\mathbf{1}_{f > 0}) d\mathbb{P}_0 \quad (39)$$

Similarly, we'll denote

$$R_{\phi,*}^\epsilon(\mathbb{P}_0, \mathbb{P}_1) = \inf_f R_\phi^\epsilon(f; \mathbb{P}_0, \mathbb{P}_1), \quad R_*^\epsilon(\mathbb{P}_0, \mathbb{P}_1) = \inf_f R^\epsilon(f; \mathbb{P}_0, \mathbb{P}_1) \qquad (40)$$

Observe that for any two sets $U_0, U_1$,

$$R_\phi^\epsilon(f; \mathbb{P}_0, \mathbb{P}_1) = R_\phi^\epsilon(f; \mathbb{P}_0|_{U_0}, \mathbb{P}_1|_{U_1}) + R_\phi^\epsilon(f; \mathbb{P}_0|_{U_0^C}, \mathbb{P}_1|_{U_1^C})$$

This decomposition reflects the fact that the adversarial surrogate risk is additive over disjoint measurable partitions of the data space. If furthermore these sets are induced by transport maps, then the optimal risks also follow this split.

**Lemma 10.** *Let $\mathbb{P}_0^*, \mathbb{P}_1^*, T_0, T_1, U_0, U_1$ and $Q$ be as in Lemma 9, and define $\mathbb{P}^* = \mathbb{P}_0^* + \mathbb{P}_1^*$, $\eta^* = d\mathbb{P}_1^*/d\mathbb{P}^*$. Then*

$$R_{\phi,*}^\epsilon(\mathbb{P}_0, \mathbb{P}_1) = R_{\phi,*}^\epsilon(\mathbb{P}_0|_{U_0}, \mathbb{P}_1|_{U_1}) + R_{\phi,*}^\epsilon(\mathbb{P}_0|_{U_0^C}, \mathbb{P}_1|_{U_1^C})$$

*and furthermore, $R_{\phi,*}^\epsilon(\mathbb{P}_0|_{U_0^C}, \mathbb{P}_1|_{U_1^C}) = \int_{Q^C} C_\phi^*(\eta^*) d\mathbb{P}^*$.*

See Appendix I.2 for a proof of Lemmas 9 and 10.

*Proof of Theorem 12.* Let $Q = \{\mathbf{x}' : \eta^*(\mathbf{x}') = 1/2\}$. Then Lemma 9 applied to $Q^C$ shows that $(\mathbb{P}_0^*|_{Q^C}, \mathbb{P}_1^*|_{Q^C})$ maximize $\bar{R}_\phi$ over $\mathcal{B}_\epsilon^\infty(\mathbb{P}_0|_{U_0}) \times \mathcal{B}_\epsilon^\infty(\mathbb{P}_1|_{U_1})$, with $U_0 = T_0^{-1}(Q^C)$ and $U_1 = T_1^{-1}(Q^C)$. Theorem 11

and Lemma 10 imply that

$$R^\epsilon(f; \mathbb{P}_0|_{U_0}, \mathbb{P}_1|_{U_1}) - R_*^\epsilon(\mathbb{P}_0|_{U_0}, \mathbb{P}_1|_{U_1}) \leq \tilde{\Phi}\left(R_\phi^\epsilon(f; \mathbb{P}_0|_{U_0}, \mathbb{P}_1|_{U_1}) - R_{\phi*}^\epsilon(\mathbb{P}_0|_{U_0}, \mathbb{P}_1|_{U_1})\right)$$
$$\leq \tilde{\Phi}\left(R_\phi^\epsilon(f; \mathbb{P}_0, \mathbb{P}_1) - R_{\phi*}^\epsilon(\mathbb{P}_0, \mathbb{P}_1)\right).$$

Next, by Lemma 10, adding $R^\epsilon(f; \mathbb{P}_0|_{U_0^C}, \mathbb{P}_1|_{U_1^C}) - R_*^\epsilon(\mathbb{P}_0|_{U_0^C}, \mathbb{P}_1|_{U_1^C})$ to both sides of the inequality above results in

$$R^\epsilon(f; \mathbb{P}_0, \mathbb{P}_1) - R_*^\epsilon(\mathbb{P}_0, \mathbb{P}_1) \leq \tilde{\Phi}\left(R_\phi^\epsilon(f; \mathbb{P}_0, \mathbb{P}_1) - R_{\phi*}^\epsilon(\mathbb{P}_0|_{U_0}, \mathbb{P}_1)\right) + R^\epsilon(f; \mathbb{P}_0|_{U_0^C}, \mathbb{P}_1|_{U_1^C}) - \int_Q C^*(\eta^*) d\mathbb{P}^*.$$

The fact that $C^*(\eta^*) = 1/2$ on $Q$ while $S_\epsilon(\mathbf{1}_{f \leq 0}) \leq 1$, $S_\epsilon(\mathbf{1}_{f>0}) \leq 1$ implies that $R^\epsilon(f; \mathbb{P}_0|_{U_0^C}, \mathbb{P}_1|_{U_1^C}) - \int_Q C^*(\eta^*) d\mathbb{P}^* \leq \frac{1}{2}\mathbb{P}(\eta^* = 1/2)$. Thus, the excess risk contribution from the region $Q$ is at most $\mathbb{P}^*(\eta^* = 1/2)/2$. $\qquad\square$

The proof of Theorem 10 follows the same steps, except that we take $Q = \{\mathbf{x}' : |\eta(\mathbf{x}') - 1/2| < \alpha\}$, see Appendix I.3 for a proof.

# 7   Related Works

**Surrogate Risk Bounds:**   The statistical consistency of surrogate risks in both the standard and adversarial context has been widely studied. Bartlett et al. (2006); Zhang (2004) establish surrogate risk bounds that apply to the class of all measurable functions while Lin (2004); Steinwart (2007) prove further results on consistency in the standard setting. Frongillo & Waggoner (2021) study the optimally of such bounds, and Bao (2023) derive bounds using the modulus of convexity of $C_\phi^*$ to construct surrogate risk bounds. Several works (Philip M. Long, 2013; Mingyuan Zhang, 2020; Awasthi et al., 2022; Mao et al., 2023a;b; Awasthi et al., 2023b) study consistency within a restricted function class; a concept known as $\mathcal{H}$-*consistency*. Mahdavi et al. (2014) combine surrogate risk bounds with surrogate generalization bounds to study the generalization of the classification error.

**Adversarial Surrogate Risk Bounds:**   Most closely related to our results are Li & Telgarsky (2023); Mao et al. (2023a). Li & Telgarsky (2023) derive a surrogate bound for convex losses in which the threshold in (10) is optimized rather than fixed at zero. Mao et al. (2023a) establish an adversarial surrogate bound for a modified $\rho$-margin loss.

**Adversarial Consistency:**   In the adversarial setting, Meunier et al. (2022); Frank & Niles-Weed (2024a) characterize which losses are adversarially consistent for all data distributions. Frank (2025) show that under reasonable distributional assumptions, a consistent loss is adversarially consistent for a specific distribution iff the adversarial Bayes classifier is unique up to degeneracy. Awasthi et al. (2021) study adversarial consistency for a well-motivated class of linear functions while Awasthi et al. (2023b); Mao et al. (2023a) study $\mathcal{H}$-consistency in the adversarial setting for specific surrogate risks. Standard and adversarial surrogate risk bounds are a central tool in the derivation of the $\mathcal{H}$-consistency bounds in this line of research. Whether the adversarial surrogate bounds presented in this paper could result in improved adversarial $\mathcal{H}$-consistency bounds remains an open problem.

**The Adversarial Bayes Classifier:**   Our proofs draw on prior work that investigates adversarial risks and adversarial Bayes classifiers. Bungert et al. (2021); Pydi & Jog (2021; 2020); Bhagoji et al. (2019); Awasthi et al. (2023a) establish existence results for the adversarial Bayes classifier, while Frank & Niles-Weed (2024b); Pydi & Jog (2020; 2021); Bhagoji et al. (2019); Frank (2025) prove minimax theorems for adversarial surrogate and classification risks. Pydi & Jog (2020) use such results to analyze the adversarial Bayes classifier, and Frank (2024) employ them to study uniqueness.

**Sample Complexity and Surrogate Risks:** The bound of Theorem 2 can be linear even for convex loss functions. For the hinge loss $\phi(\alpha) = \max(1 - \alpha, 0)$, the function $\phi$ computes to $\phi(\theta) = |\theta|$. Mahdavi et al. (2014) emphasize the importance of a linear convergence rate in a surrogate risk bound. They note that convex surrogates with favorable sample complexity often fail to satisfy strong surrogate risk bounds, due to Theorem 2 Frongillo & Waggoner (2021): convex losses which are locally strictly convex and Lipschitz achieve at best a square root surrogate risk rate. Thus, Proposition 1 suggests that favorable sample complexity guarantees for convex surrogates may require distributional conditions such as Massart's noise condition, under which Massart & Nédélec (2006) also show improved sample complexity.

## 8 Conclusion

In conclusion, we prove surrogate risk bounds for adversarial risks. When $\phi$ is adversarially consistent or the distribution of optimal adversarial attacks satisfies Massart's noise condition, we obtain a linear surrogate risk bound. In the general case, we prove a concave distribution-dependent bound. Understanding the optimality of the concave bound remains an open problem, as does understanding how these bounds interact with the sample complexity of estimating the surrogate risk. While related questions have been studied in the standard setting (Frongillo & Waggoner, 2021; Mahdavi et al., 2014), the adversarial context remains largely unexplored. Advancing these directions could bridge the current gap between theoretical guarantees and practical robustness in adversarial learning.

### Acknowledgments

Natalie Frank was supported in part by the Research Training Group in Modeling and Simulation funded by the National Science Foundation via grant RTG/DMS – 1646339, NSF grant DMS-2210583, and NSF TRIPODS II - DMS 2023166.

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
