## Contents of Appendix

## A  Proof of Theorem 1

**Lemma 11.** *Assume $\phi$ is consistent. Then $C_\phi^*(\eta) = \phi(0)$ implies that $\eta = 1/2$.*

This result appeared as Lemma 7 of Frank (2025).

*Proof.* If $\phi$ is consistent and 0 minimizes $C_\phi(\eta, \alpha)$, then 0 must also minimize $C(\eta, \alpha) = \eta \mathbf{1}_{\alpha \leq 0} + (1-\eta)\mathbf{1}_{\alpha > 0}$ and consequently $\eta \leq 1/2$. However $C_\phi(\eta, \alpha) = C_\phi(1-\eta, -\alpha)$ so that 0 must minimize $C(1-\eta, -\alpha)$ as well. Consequently, $1 - \eta \leq 1/2$ and thus $\eta$ must actually equal $1/2$. $\qquad\square$

*Proof of Theorem 1.* **Forward direction:**  Assume that $\phi$ is consistent. Note that $C_\phi^*(\eta) \leq C_\phi(\eta, 0) = \phi(0)$ for any $\eta$. Thus Lemma 11 implies that $C_\phi^*(\eta) < \phi(0)$ for $\eta \neq 1/2$.

**Backward direction:**  Assume that $C_\phi^*(\eta) < \phi(0)$ for all $\eta \neq 1/2$. Notice that if $\eta = 1/2$, $C(1/2, \alpha)$ is constant in $\alpha$ so *any* sequence $\alpha_n$ minimizes $C(1/2, \cdot)$. We will show if $\eta > 1/2$ and $\alpha_n$ is a minimizing sequence of $C_\phi(\eta, \cdot)$, then $\alpha_n > 0$ for sufficiently large $n$, and thus must also minimize $C(\eta, \cdot)$. An analogous argument will imply that if $\eta < 1/2$, any minimizing sequence of $C_\phi(\eta, \cdot)$ must also minimize $C(\eta, \cdot)$ as well.

Assume $\eta > 1/2$ and let $\alpha_n$ be any minimizing sequence of $C_\phi(\eta, \cdot)$. Let $\alpha^*$ be a limit point of the sequence $\alpha_n$ in the extended real number line $\overline{\overline{\mathbb{R}}}$. Then $\alpha^*$ is a minimizer of $C_\phi(\eta, \alpha)$. Next, observe that one of $\phi(\alpha^*)$, $\phi(-\alpha^*)$ is larger that or equal to $\phi(0)$ and the other is less than or equal to $\phi(0)$. As $\eta > 1/2$ and $\alpha^*$ is a minimizer of $C_\phi(\eta, \cdot)$ and $C_\phi(\eta, \alpha^*) < \phi(0)$, one can conclude that $\phi(\alpha^*) < \phi(0)$ and consequently $\alpha^* > 0$.

Therefore, every limit point of the sequence $\{\alpha_n\}$ is strictly positive. Consequently, one can conclude that $\alpha_n > 0$ for sufficiently large $n$.

$\qquad\square$

## B  Linear Surrogate Risk Bounds—Proof of Proposition 1

In this appendix, we will find it useful to study the function

$$C_\phi^-(\eta) = \inf_{z(2\eta - 1) \leq 0} C_\phi(\eta, z)$$

introduced by Bartlett et al. (2006). This function maps $\eta$ to the smallest value of the conditional $\phi$-risk assuming an incorrect classification. The symmetry $C_\phi(\eta, \alpha) = C_\phi(1-\eta, -\alpha)$ implies $C_\phi^-(\eta) = C_\phi^-(1-\eta)$. Further, the function $C_\phi^-$ is concave on each of the intervals $[0, 1/2]$ and $[1/2, 1]$, as it is an infimum of linear functions on each of these regions. The next result examines the monotonicity properties of $C_\phi^*$ and $C_\phi^-$.

**Lemma 12.** *The function $C_\phi^*$ is non-decreasing on $[0, 1/2]$ and non-increasing on $[1/2, 1]$. In contrast, $C_\phi^-$ is non-increasing on $[0, 1/2]$ and non-decreasing on $[1/2, 1]$*

*Proof.* The symmetry $C_\phi^*(\eta) = C_\phi^*(1-\eta)$ and $C_\phi^-(\eta) = C_\phi^-(1-\eta)$ implies that it suffices to check monotonicity on $[0, 1/2]$. Observe that

$$C_\phi(\eta, \alpha) - C_\phi(\eta, -\alpha) = \eta(\phi(\alpha) - \phi(-\alpha)) + (1-\eta)(\phi(-\alpha) - \phi(\alpha)) = (2\eta - 1)(\phi(\alpha) - \phi(-\alpha)).$$

If $\eta \leq 1/2$, then this quantity is non-negative when $\alpha \leq 0$. Therefore, when computing $C_\phi^*$ over $[0, 1/2]$, it suffices to minimize $C_\phi(\eta, \alpha)$ over $\alpha \leq 0$. In other words, for $\eta \leq 1/2$,

$$C_\phi^*(\eta) = \inf_\alpha C_\phi(\eta, \alpha) = \inf_{\alpha \leq 0} C_\phi(\eta, \alpha)$$

For any fixed $\alpha \leq 0$, the quantity $C_\phi(\eta, \alpha)$ is non-increasing in $\eta$ and thus $C_\phi^*(\eta_1) \leq C_\phi^*(\eta_2)$ when $\eta_1 \leq \eta_2 \leq 1/2$.

In contrast, for any $\alpha \geq 0$, the quantity $C_\phi(\eta, \alpha)$ is non-decreasing in $\eta$ and thus $C_\phi^-(\eta_1) \geq C_\phi^-(\eta_2)$ when $\eta_1 \leq \eta_2 \leq 1/2$.

$\qquad\square$

Next we'll prove a useful lower bound on $C_\phi^-$.

**Lemma 13.** *For all $\eta \in [0, 1]$,*

$$C_\phi^-(\eta) \geq |1 - 2\eta|\phi(0) + 2\min(\eta, 1 - \eta)C_\phi^*(\eta) \tag{41}$$

*Proof.* First, assume that $\eta \leq 1/2$ and observe that $\eta$ is the convex combination $\eta = 2\eta \cdot 1/2 + (1 - 2\eta) \cdot 0$. By the concavity of $C_\phi^-$ on $[0, 1/2]$,

$$C_\phi^-(\eta) = C_\phi^-\left(2\eta \cdot \frac{1}{2} + (1 - 2\eta) \cdot 0\right) \geq (1 - 2\eta)C_\phi^-(0) + 2\eta C_\phi^-\left(\tfrac{1}{2}\right)$$

However, $C_\phi^-(0) = \phi(0)$ while $C_\phi^-(1/2) = C_\phi^*(1/2)$. Further, Lemma 12 implies that $C_\phi^*(1/2) \geq C_\phi^*(\eta)$, yielding the inequality

$$C_\phi^-(\eta) \geq (1 - 2\eta)\phi(0) + 2\eta C_\phi^*(\eta)$$

Symmetry $C_\phi^-(\eta) = C_\phi^-(1 - \eta)$ then implies (41).

$\square$

*Proof of Proposition 1.* If $C(\eta, f) - C^*(\eta) = 0$ then (8) holds trivially. Otherwise, $C(\eta, f) - C^*(\eta) = |2\eta - 1|$. If $C(\eta, f) = |2\eta - 1|$, then

$$
\begin{aligned}
C(\eta, f) - C^*(\eta) = |2\eta - 1| &= |2\eta - 1| \cdot \frac{\phi(0) - C_\phi^*(\eta)}{\phi(0) - C_\phi^*(\eta)} \\
&\leq \frac{1}{\phi(0) - C_\phi^*(\eta)}\left(\left(|2\eta - 1|\phi(0) + (1 - |2\eta - 1|)C_\phi^*(\eta)\right) - C_\phi^*(\eta)\right)
\end{aligned}
\tag{42}
$$

At the same time, because $|\eta - 1/2| \geq \alpha$ $\mathbb{P}$-a.e. Lemma 12 implies that $C_\phi^*(\eta) \leq C_\phi^*(1/2 - \alpha)$ $\mathbb{P}$-a.e. Furthermore, the relation $2\min(\eta, 1 - \eta) = 1 - |1 - 2\eta|$ together with (41) shows that

$$|2\eta - 1|\phi(0) + (1 - |2\eta - 1|)C_\phi^*(\eta) \leq C_\phi^-(\eta).$$

Therefore, (42) is bounded above by

$$\leq \frac{1}{\phi(0) - C_\phi^*\left(\frac{1}{2} - \alpha\right)}\left(C_\phi^-(\eta) - C_\phi^*(\eta)\right) \leq \frac{1}{\phi(0) - C_\phi^*\left(\frac{1}{2} - \alpha\right)}\left(C_\phi(\eta, f) - C_\phi^*(\eta)\right). \tag{43}$$

The last equality follows from the supposition $C(\eta, f) - C^*(\eta) = |2\eta - 1|$, as it implies $(2\eta - 1)f \leq 0$, and thus $C_\phi(\eta, f) \geq C_\phi^-(\eta)$. Consequently, (43) implies (8).

Integrating (8) with respect to $\mathbb{P}$ then produces the surrogate bound (9).

$\square$

## C Proof of Lemma 1

*Proof of Lemma 1.* If $\mathbf{x}' \in \overline{B_\epsilon(\mathbf{x})}$ then $S_\epsilon(g)(\mathbf{x}) \geq g(\mathbf{x}')$. Thus if $\gamma$ is a coupling between $\mathbb{Q}$ and $\mathbb{Q}'$ supported on $\Delta_\epsilon$, then $S_\epsilon(g)(\mathbf{x}) \geq g(\mathbf{x}')$ $\gamma$-a.e. Integrating this inequality in $\gamma$ produces

$$\int S_\epsilon(g)d\mathbb{Q} \geq \int g d\mathbb{Q}'.$$

Taking the supreumum over all $\mathbb{Q} \in \mathcal{B}_\epsilon^\infty(\mathbb{Q})$ then proves the result. $\square$

# D  Proof of Item 1), Theorem 8

We will work with an alternative primal problem from Frank & Niles-Weed (2024b) that will make it easier to study the dual. Consider minimizing

$$\Theta(h_0, h_1) = \int S_\epsilon(h_1) d\mathbb{P}_1 + \int S_\epsilon(h_0) d\mathbb{P}_0$$

over the convex set

$$S_\phi = \left\{ \begin{array}{l} (h_0, h_1) \colon h_0, h_1 \colon K^\epsilon \to \overline{\mathbb{R}} \text{ Borel}, 0 \leq h_0, h_1 \text{ and for} \\ \text{all } \mathbf{x} \in \mathbb{R}^d, \eta \in [0,1], \eta h_1(\mathbf{x}) + (1-\eta) h_0(\mathbf{x}) \geq C_\phi^*(\eta) \end{array} \right\} \tag{44}$$

Then strong duality holds with $\Theta$ in place of $R_\phi^\epsilon$. Furthermore, there exist minimizers over the set of $\overline{\mathbb{R}}$-valued functions, where $\overline{\mathbb{R}} = \mathbb{R} \cup \{-\infty, +\infty\}$.

**Theorem 13.** *Define $\bar{R}_\phi$ as in (13).*

$$\inf_{(h_0, h_1) \in S_\phi} \Theta(h_0, h_1) = \sup_{\substack{\mathbb{P}_0' \in \mathcal{B}_\epsilon^\infty(\mathbb{P}_0) \\ \mathbb{P}_1' \in \mathcal{B}_\epsilon^\infty(\mathbb{P}_1)}} \bar{R}_\phi(\mathbb{P}_0', \mathbb{P}_1')$$

*Furthermore, the infimum is attained at some $\overline{\mathbb{R}}$-valued $h_0^*$, $h_1^*$.*

See (Frank & Niles-Weed, 2024b, Lemma 14, Lemma 21) for a proof of this result. Theorem 7 already implies that the dual problem attains its supremum. Complimentary slackness conditions further characterize minimizers and maximizers.

**Theorem 14** (Complementary Slackness)**.** *The pair $(h_0^*, h_1^*)$ minimize $\Theta$ over $S_\phi$ and the measures $(\mathbb{P}_0^*, \mathbb{P}_1^*)$ maximize $\bar{R}_\phi$ over $\mathcal{B}_\epsilon^\infty(\mathbb{P}_0) \times \mathcal{B}_\epsilon^\infty(\mathbb{P}_1)$ iff the following two conditions hold:*

*1)*

$$\int S_\epsilon(h_1^*) d\mathbb{P}_1 = \int h_1^* d\mathbb{P}_1^* \quad and \quad \int S_\epsilon(h_0^*) d\mathbb{P}_0 = \int h_0^* d\mathbb{P}_0^*$$

*2)*

$$\eta^* h_1^* + (1 - \eta^*) h_0^* = C_\phi^*(\eta^*) \quad \mathbb{P}^*\text{-}a.e.$$

See Frank & Niles-Weed (2024b, Lemma 15) for a proof. Theorems 13 and 14 apply to the conditional risk $C^*(\eta)$ as $C^*(\eta) = C_\phi^*(\eta)$ for the hinge $\phi(\alpha) = \frac{1}{2}(1-\alpha)_+$.

We will use a characterization of consistency similar to Theorem 1 in the proof of Item 1), Theorem 8.

**Theorem 15.** *A loss function $\phi$ is consistent iff $C_\phi^*(\eta)$ has a strict maximum at $1/2$.*

*Proof.* If $C_\phi^*(1/2) = \phi(0)$, this statement is exactly Theorem 1. If $C_\phi^*(1/2) < \phi(0)$, Frank & Niles-Weed (2024a, Proposition 3) implies that $\phi$ is consistent. It remains to show that if $C_\phi^*(1/2) < \phi(0)$, then $C_\phi^*(\eta)$ has a strict maximum at $1/2$. As every sequence has a convergent subsequence in $\overline{\mathbb{R}}$, one can assume that $C_\phi(1/2, \cdot)$ has a minimizer $\alpha^*$ and $C_\phi^*(1/2) < \phi(0)$ implies $\alpha^* \neq 0$. Symmetry of $C_\phi(1/2, \cdot)$ implies that we can assume $\alpha^* > 0$, and thus $\phi(\alpha^*) \leq \phi(0)$ and $\phi(-\alpha^*) \geq \phi(0)$. The fact that $C_\phi^*(1/2, \alpha^*) < \phi(0)$ implies that in fact $\phi(\alpha^*) < \phi(0) \leq \phi(-\alpha^*)$. Next, observe that for any $\alpha$,

$$C_\phi(\eta, \alpha) = \frac{1}{2}(\phi(\alpha) + \phi(-\alpha)) + (\eta - \frac{1}{2})(\phi(\alpha) - \phi(-\alpha))$$

Thus, one can bound $C_\phi^*(\eta)$ by

$$C_\phi^*(\eta) \leq C_\phi(\eta, \alpha^*) = \frac{1}{2}(\phi(\alpha^*) + \phi(-\alpha^*)) + (\eta - 1/2)(\phi(\alpha^*) - \phi(-\alpha^*)) = C_\phi^*(1/2) + (\eta - 1/2)(\phi(\alpha^*) - \phi(-\alpha^*))$$

Thus if $\eta > 1/2$, then $C_\phi^*(\eta) < C_\phi^*(1/2)$. Symmetry implies that $C_\phi^*(\eta) < C_\phi^*(1/2)$ for all $\eta$. Thus $C_\phi^*$ has a strict maximum at $1/2$.

$\square$

Next, Theorem 14 implies that minimizers of $\Theta$ assume their suprema. This observation will make it easier to work with these functions.

**Lemma 14.** *If $(h_0^*, h_1^*)$ minimizes $\Theta$ over $S_\phi$, then the functions $h_0^*$, $h_1^*$ assume their suprema $\mathbb{P}_0$-a.e. and $\mathbb{P}_1$-a.e. respectively*

*Proof.* We will show the statement for $h_1^*$, the argument for $h_0^*$ is analogous. Let $\gamma_1^*$ be the coupling between $\mathbb{P}_1$ and $\mathbb{P}_1^*$ that achieves the minimum $W_\infty$ distance. Lemma 1 and Item 1) of Theorem 14 implies that

$$S_\epsilon(h_1)(\mathbf{x}) = h_1(\mathbf{x}') \quad \gamma_1^*\text{-a.e.}$$

and thus $h_1^*$ assumes its maximum over closed $\epsilon$-balls $\mathbb{P}_1^*$-a.e. $\square$

**Lemma 15.** *If $(h_0^*, h_1^*) \in S_\phi$, then at any $\mathbf{x}$ either $h_1^*(\mathbf{x}) \geq C_\phi^*(\frac{1}{2})$ or $h_0^*(\mathbf{x}) > C_\phi^*(\frac{1}{2})$.*

*Proof.* If $(h_0^*, h_1^*) \in S_\phi$, then at any point $\mathbf{x}$,

$$\frac{1}{2} h_0^*(\mathbf{x}) + \frac{1}{2} h_1^*(\mathbf{x}) \geq C_\phi^*(\frac{1}{2}).$$

The inequality $h_0^*(\mathbf{x}) \leq C_\phi^*(1/2)$ implies $h_1^*(\mathbf{x}) \geq C_\phi^*(1/2)$. Thus either $h_0^*(\mathbf{x}) > C_\phi^*(1/2)$ or $h_1^*(\mathbf{x}) \geq C_\phi^*(1/2)$ at any point. $\square$

*Proof of Item 1) of Theorem 8 .* Let $\phi_{\text{hinge}}(\alpha) = \frac{1}{2}(1-\alpha)_+$, then $C_{\phi_{\text{hinge}}}^*(\eta) = C^*(\eta)$.

Let $(h_0^*, h_1^*)$ minimize $\Theta$ over $S_\phi$ and $(\mathbb{P}_0^*, \mathbb{P}_1^*)$ maximize $\bar{R}_\phi$ over $\mathcal{B}_\epsilon^\infty(\mathbb{P}_0) \times \mathcal{B}_\epsilon^\infty(\mathbb{P}_1)$. We will show that the functions defined by

$$\tilde{h}_1^*(\mathbf{x}) = \mathbf{1}_{h_1^*(\mathbf{x}) \geq C_\phi^*(\frac{1}{2})} \quad \tilde{h}_0^*(\mathbf{x}) = \mathbf{1}_{h_0^*(\mathbf{x}) > C_\phi^*(\frac{1}{2})}$$

maximize $\Theta$ over $S_{\phi_{\text{hinge}}}$ and $(\mathbb{P}_0^*, \mathbb{P}_1^*)$ maximize $\bar{R}_{\phi_{\text{hinge}}}$ by verifying the constraint $(\tilde{h}_0^*, \tilde{h}_1^*) \in S_{\phi_{\text{hinge}}}$ and the complimentary slackness conditions. The proof thus consists of three steps: verifying $(\tilde{h}_0^*, \tilde{h}_1^*) \in S_{\phi_{\text{hinge}}}$, and checking the two complementary slackness conditions in Theorem 14.

1) **Verifying the constraint defining $S_{\phi_{\text{hinge}}}$:** Observe that Lemma 15 implies that at any $\mathbf{x}$, at least one of $\tilde{h}_0^*(\mathbf{x})$ and $\tilde{h}_1^*(\mathbf{x})$ is 1, and thus

$$\eta h_1^*(\mathbf{x}) + (1-\eta) h_0^*(\mathbf{x}) \geq \min(\eta, 1-\eta) = C_{\phi_{\text{hinge}}}^*(\eta)$$

2) **Verifying Item 1) of Theorem 14:** Observe that Lemma 14 implies that $S_\epsilon(\mathbf{1}_{h_1^* \geq C_\phi^*(1/2)})(\mathbf{x}) = \mathbf{1}_{S_\epsilon(h_1^*)(\mathbf{x}) \geq C_\phi^*(1/2)}$ $\mathbb{P}_1^*$-a.e. Subsequently, the Item 1) of Theorem 14 implies that

$$S_\epsilon(\mathbf{1}_{h_1^* \geq C_\phi^*(1/2)})(\mathbf{x}) = \mathbf{1}_{h_1^*(\mathbf{x}') \geq C_\phi^*(1/2)} \quad \gamma_1^*\text{-a.e.,}$$

verifying the first complimentary slackness condition for $\tilde{h}_1^*$. Analogous reasoning shows that

$$S_\epsilon(\mathbf{1}_{h_0^* > C_\phi^*(1/2)})(\mathbf{x}) = \mathbf{1}_{h_0^*(\mathbf{x}') > C_\phi^*(1/2)} \quad \gamma_0^*\text{-a.e.}$$

3) **Verifying Item 2) of Theorem 14:** Theorem 14 implies that $\eta^* h_1^*(\mathbf{x}') + (1-\eta^*) h_0^*(\mathbf{x}') = C_\phi^*(\eta^*) \leq C_\phi^*(1/2)$, and thus Lemma 15 implies that *exactly* one of $h_1^*(\mathbf{x}')$ and $h_0^*(\mathbf{x}')$ equals 1 and the other equals 0. We'll consider the cases $\eta^*(\mathbf{x}') < 1/2$, $\eta^*(\mathbf{x}') = 1/2$, and $\eta^*(\mathbf{x}') > 1/2$ separately. In these three separate cases, we will explicitly use the formula $C_{\phi_{\text{hinge}}}^*(\eta) = \min(\eta, 1-\eta)$.

**When $\eta^*(\mathbf{x}') = 1/2$:** As exactly one of $h_0^*(\mathbf{x}')$ and $h_1^*(\mathbf{x}')$ is 1:

$$\frac{1}{2}\tilde{h}_0^*(\mathbf{x}') + \frac{1}{2}\tilde{h}_1^*(\mathbf{x}') = \frac{1}{2} = C^*_{\phi_{\text{hinge}}}\left(\frac{1}{2}\right)$$

**When $\eta^*(\mathbf{x}') < 1/2$:** Observe that if $h_0^*(\mathbf{x}') > h_1^*(\mathbf{x}')$, then

$$\eta^* h_0^*(\mathbf{x}') + (1 - \eta^*)h_1^*(\mathbf{x}') < \eta^* h_1^*(\mathbf{x}') + (1 - \eta^*)h_0^*(\mathbf{x}') = C^*_\phi(\eta^*),$$

which would violate the constraint on $S_\phi$. Therefore, $h_0^*(\mathbf{x}') \leq h_1^*(\mathbf{x}')$. Next, Theorem 15 implies that $\eta^* h_1^*(\mathbf{x}') + (1 - \eta^*)h_0^*(\mathbf{x}') = C^*_\phi(\eta^*) < C^*_\phi(1/2)$. These two statements together with Lemma 15 imply that $h_0^*(\mathbf{x}') < C^*_\phi(1/2)$ and $h_1^*(\mathbf{x}') \geq C^*_\phi(1/2)$. However, $h_0^*(\mathbf{x}') = C^*_\phi(1/2)$ would still violate $\eta^* h_1^*(\mathbf{x}') + (1 - \eta^*)h_0^*(\mathbf{x}') < C^*_\phi(1/2)$ and therefore, $h_0^*(\mathbf{x}') < C^*_\phi(1/2)$. Therefore,

$$\eta^* \tilde{h}_1^* + (1 - \eta^*)\tilde{h}_0^* = \eta^* = C^*_{\phi_{\text{hinge}}}(\eta^*)$$

**When $\eta^*(\mathbf{x}') > 1/2$:** Argument is analogous to the previous case.

$\square$

# E   Proof of Lemma 7

We define the *concave conjugate* of a function $h$ as

$$h_*(y) = \inf_{x \in \text{dom}(h)} yx - h(x)$$

Recall that $\text{conc}(h)$ as defined in (15) is the biconjugate $h_{**}$. Consequently, $\text{conc}(h)$ can be expressed as

$$\text{conc}(h)(x) = \inf\{\ell(x) : \ell \text{ linear, and } \ell \geq h \text{ on } \text{dom}(h)\} \tag{45}$$

Lemma 7 is a consequence of the properties of concave conjugates.

**Lemma 16.** *Let $h : [a, b] \to \mathbb{R}$ be a non-decreasing function. Then $\text{conc}(h)$ is non-decreasing as well.*

*Proof.* We will argue that if $h$ is non-decreasing, then it suffices to consider the infimum in (45) over non-decreasing linear functions. Observe that if $\ell$ is a decreasing linear function with $\ell(x) \geq h(x)$ then the constant function $\ell(b)$ satisfies

$$\ell(x) \geq \ell(b) \geq h(b) \geq h(x)$$

for any $x \in [a, b]$. Therefore,

$$\text{conc}(h)(x) = \inf\{\ell(x) : \ell \text{ linear, non-decreasing, and } \ell \geq h\}$$

$\square$

**Lemma 17.** *Let $h : [0, b] \to \mathbb{R}$ be a non-decreasing function that is right-continuous at zero with $h(0) = 0$. Then $\sup_y h_*(y) = 0$. Furthermore, there is a sequence $y_n$ with $y_n \to \infty$ and $\lim_{n\to\infty} h_*(y_n) = 0$.*

*Proof.* First, notice that

$$h_*(y) = \inf_{x \in [0,b]} yx - h(x) \leq y \cdot 0 - h(0) = 0 \tag{46}$$

for any $y \in \mathbb{R}$. It remains to show a sequence $y_n$ for which $\lim_{n\to\infty} h_*(y_n) = 0$.

We will argue than any sequence $y_n$ with

$$y_n > nh(b) \geq \sup_{x \in [1/n,b]} \frac{h(x)}{x} \tag{47}$$

satisfies this property.

If $x \in [1/n, b]$ and $y_n$ satisfies (47) then

$$xy_n - h(x) = x\left(y_n - \frac{h(x)}{x}\right) > 0$$

and thus (46) implies that

$$h_*(y_n) = \inf_{x \in [0, 1/n)} xy_n - h(x)$$

The monononicity of $h$ then implies that

$$h_*(y_n) \geq -h(1/n)$$

and

$$\lim_{n \to \infty} h_*(y_n) \geq 0$$

because $h$ is right-continuous at zero. This relation together with (46) implies the result.

$\square$

*Proof of Lemma 7.* Lemma 16 implies that $\text{conc}(h)$ is non-decreasing. Standard results in convex analysis imply that $\text{conc}(h)$ is continuous on $(0, 1/2)$ (Hiriart-Urruty & Lemaréchal, 2001, Lemma 3.1.1) and upper semi-continuous on $[0, 1/2]$ (Hiriart-Urruty & Lemaréchal, 2001, Theorem 1.3.5). Thus monotonicity implies that for all $x \in [0, 1/2]$, $\text{conc}(h)(x) \leq \text{conc}(h)(1/2)$ and thus $\lim_{x \to 1/2} \text{conc}(h)(x) \leq \text{conc}(h)(1/2)$. We will show the opposite inequality, implying that $\text{conc}\, h$ is continuous at $1/2$.

First, as the constant function $h(1/2)$ is an upper bound on $h$, one can conclude that $\text{conc}(h)(1/2) = h(1/2) = 1$. Next, recall that $\text{conc}(h)$ can be expressed as an infimum of linear functions as in (45). If $\ell \geq h$, then $\ell(0) \geq 0$ and $\ell(1/2) \geq 1$. Therefore,

$$\ell(\tfrac{1}{2} - \delta) = \ell((1 - 2\delta) \cdot \tfrac{1}{2} + 2\delta \cdot 0) = (1 - 2\delta)\ell(\tfrac{1}{2}) + 2\delta\ell(0) \geq 1 - 2\delta.$$

Therefore, the representation (45) implies that $\text{conc}(h)(1/2 - \delta) \geq 1 - 2\delta$. Taking $\delta \to 0$ proves that $\lim_{x \to 1/2} \text{conc}(h)(x) \geq 1$. Thus, $\text{conc}(h)$ is continuous at $1/2$, if viewed as a function on $[0, 1/2]$.

Next, Lemma 17 implies that $h_{**}(0) = 0$:

$$h_{**}(0) = \inf_{y \in \mathbb{R}} -h_*(y) = -\sup_{y \in \mathbb{R}} h_*(y) = 0.$$

Finally, it remains to show that $h_{**}$ is continuous at 0. The monotonicity of $h_{**}$ implies that $\lim_{y \to 0^+} h_{**}(y) = \inf_{y \in (0, 1/2]} h_{**}(y)$ and consequently

$$\lim_{y \to 0^+} h_{**}(y) = \inf_{y \in (0, 1/2]} \inf_{x \in \mathbb{R}} yx - h_*(x) = \inf_{x \in \mathbb{R}} \inf_{y \in (0, 1/2]} yx - h_*(x) = \inf_{x \in \mathbb{R}} -h_*(x) + \begin{cases} 0 & \text{if } x \geq 0 \\ \frac{x}{2} & \text{if } x < 0 \end{cases}$$

$$= \min\left(\inf_{x \geq 0} -h_*(x), \inf_{x < 0} \frac{x}{2} - h_*(x)\right) \tag{48}$$

However, Lemma 17 implies that

$$\inf_{x \geq 0} -h_*(x) = \inf_{x \in \mathbb{R}} -h_*(x) = 0 \tag{49}$$

Notice that if $x \leq 0$,

$$h_*(x) = \inf_{z \in [0, 1/2]} xz - h(z) = \frac{x}{2} - h\left(\frac{1}{2}\right) = \frac{x}{2} - 1 \tag{50}$$

Consequently, (49) and (50) implies that (48) evaluates to 0.

$\square$

# F   Proof of Proposition 2

A modified version of Jensen's inequality will be used at several points in the proof of Proposition 2.

**Lemma 18.** *Let $G$ be a concave function with $G(0) = 0$ and let $\nu$ be a measure with $\nu(\mathbb{R}^d) \leq 1$. Then*

$$\int G(f) d\nu \leq G\left(\int f d\nu\right)$$

*Proof.* The inequality trivially holds if $\nu(\mathbb{R}^d) = 0$, so we assume $\nu(\mathbb{R}^d) > 0$. Jensen's inequality implies that

$$\int G(f) d\nu = \nu(\mathbb{R}^d)\left(\frac{1}{\nu(\mathbb{R}^d)}\int G(f) d\nu\right) \leq \nu(\mathbb{R}^d)G\left(\frac{1}{\nu(\mathbb{R}^d)}\int f d\nu\right).$$

As $G(0) = 0$, concavity implies that

$$\nu(\mathbb{R}^d)G\left(\frac{1}{\nu(\mathbb{R}^d)}\int f d\nu\right) = \nu(\mathbb{R}^d)G\left(\frac{1}{\nu(\mathbb{R}^d)}\int f d\nu\right) + (1 - \nu(\mathbb{R}^d)G(0) \leq G\left(\int f d\nu\right)$$

$\square$

To facilitate the application of Jensen's inequality, the proof will be carried out using integrated quantities. Let $\mathbb{P}_0^*, \mathbb{P}_1^*$ be any maximizers of $\bar{R}_\phi$, which also maximize $\bar{R}$ by Theorem 8. Set $\mathbb{P}^* = \mathbb{P}_0^* + \mathbb{P}_1^*$, $\eta^* = d\mathbb{P}_1^*/d\mathbb{P}^*$. Define

*Proof of Proposition 2.* Let $\gamma_0^*$, $\gamma_1^*$ be the couplings between $\mathbb{P}_0$, $\mathbb{P}_0^*$ and $\mathbb{P}_1$, $\mathbb{P}_1^*$ respectively that achieve the infimum in (11). Define $I_1(f)$, $I_0(f)$, $I_1^\phi(f)$, and $I_0^\phi(f)$ by

$$I_1(f) = \int i_1(f) d\gamma_1^*, \quad I_1^\phi(f) = \int i_1^\phi(f) d\gamma_1^*, \quad I_0(f) = \int i_0(f) d\gamma_0^*, \quad I_0^\phi(f) = \int i_0^\phi(f) d\gamma_0^*.$$

We will prove 
$$I_0(f) \leq \frac{1}{2}\tilde{\Phi}\big(2I_0^\phi(f)\big). \tag{51}$$
$$I_1(f) \leq \frac{1}{2}\tilde{\Phi}\big(2I_1^\phi(f)\big) \tag{52}$$

The concavity of $\tilde{\Phi}$ then implies that

$$R^\epsilon(f) - R_*^\epsilon = I_1(f) + I_0(f) \leq \frac{1}{2}\tilde{\Phi}\big(2I_1^\phi(f)\big) + \frac{1}{2}\tilde{\Phi}\big(2I_0^\phi(f)\big) \leq \tilde{\Phi}\Big(\frac{1}{2}2I_1^\phi(f) + \frac{1}{2}2I_0^\phi(f)\Big) = \tilde{\Phi}\big(R_\phi^\epsilon(f) - R_{\phi,*}^\epsilon\big).$$

We will prove (52), the argument for (51) is analogous. Next, let $\gamma_1^*$ be the coupling between $\mathbb{P}_1$ and $\mathbb{P}_1^*$ supported on $\Delta_\epsilon$. The assumption on $\Phi$ implies that

$$C(\eta^*(\mathbf{x}'), f(\mathbf{x}')) - C^*(\eta^*(\mathbf{x}')) \leq \Phi\big(C_\phi(\eta^*(\mathbf{x}'), f(\mathbf{x}')) - C_\phi^*(\eta^*(\mathbf{x}'))\big) \tag{53}$$

and consequently,

$$\int C(\eta^*(\mathbf{x}'), f(\mathbf{x}')) - C^*(\eta^*(\mathbf{x}'))d\gamma_1^* \leq \Phi\left(\int C_\phi(\eta^*(\mathbf{x}'), f(\mathbf{x}')) - C_\phi^*(\eta^*(\mathbf{x}'))d\gamma_1^*\right) \leq \Phi(I_1^\phi(f)). \tag{54}$$

To bound the term $S_\epsilon(\mathbf{1}_{f\leq 0})(\mathbf{x}) - \mathbf{1}_{f(\mathbf{x}')\leq 0}$, we consider two different cases for $(\mathbf{x}, \mathbf{x}')$. Define the sets $D_1$, $E_1$ as in (25), (26). We will show that if $T_1$ is any of the sets $D_1$, $E_1$, then

$$\int_{T_1} S_\epsilon(\mathbf{1}_{f\leq 0})(\mathbf{x}) - \mathbf{1}_{f(\mathbf{x}')\leq 0}d\gamma_1^*$$

$$\leq \left(\int \frac{1}{G\big((\phi(0) - C_\phi^*(\eta^*(\mathbf{x})))/2\big)}d\gamma_1^*\right)^{\frac{1}{2}} G\left(\int_{T_1}\big((S_\epsilon(\phi \circ f)(\mathbf{x}) - \phi(f(\mathbf{x}'))) + (C_\phi(\eta^*(\mathbf{x}'), f(\mathbf{x}')) - C_\phi^*(\eta^*(\mathbf{x}')))\,d\gamma_1^*\right)^{\frac{1}{2}}$$

$$\tag{55}$$

Thus because $G$ is concave and non-decreasing, the composition $\sqrt{G}$ is as well. Thus summing the inequality (55) over $T_1 \in \{D_1, E_1\}$ results in

$$\int S_\epsilon(\mathbf{1}_{f \leq 0})(\mathbf{x}) - \mathbf{1}_{f(\mathbf{x}') \leq 0} d\gamma_1^* \leq 2 \left( \int \frac{1}{G\left(\phi(0) - C_\phi^*(\eta^*(\mathbf{x}'))\right)} d\mathbb{P}^* \right)^{\frac{1}{2}} G\left(\frac{1}{2} I_1^\phi(f)\right)^{\frac{1}{2}} \tag{56}$$

Summing (54) and (56) results in (52).

It remains to show the inequality (55) for the two sets $D_1, E_1$.

A) **On the set $D_1$:**

If $S_\epsilon(\mathbf{1}_{f \leq 0})(\mathbf{x}) = \mathbf{1}_{f(\mathbf{x}') \leq 0}$, then $\int_{D_1} S_\epsilon(\mathbf{1}_{f \leq 0})(\mathbf{x}) - \mathbf{1}_{f(\mathbf{x}') \leq 0} d\gamma_1^* = 0$ while the left-hand side of (55) is non-negative by Lemma 1, which implies (55) for $T_1 = D_1$.

B) **On the set $E_1$:**

Lemma 1 then implies that $S_\epsilon(\phi \circ f)(\mathbf{x}) - \phi(f(\mathbf{x}')) \geq 0$ $\gamma_1^*$-a.e. and thus Lemma 5 implies

$$S_\epsilon(\mathbf{1}_{f \leq 0})(\mathbf{x}) - \mathbf{1}_{f(\mathbf{x}') \leq 0} = 1 = \frac{\sqrt{G\left(\phi(0) - C_\phi^*(\eta^*(\mathbf{x}'))\right)}}{\sqrt{G\left(\phi(0) - C_\phi^*(\eta^*(\mathbf{x}'))\right)}} \leq \frac{\sqrt{G\left(S_\epsilon(\phi \circ f)(\mathbf{x}) - \phi(f(\mathbf{x}'))\right)}}{\sqrt{G\left(\phi(0) - C_\phi^*(\eta^*(\mathbf{x}'))\right)}} \quad \gamma_1^*\text{-a.e.} \tag{57}$$

Now the Cauchy-Schwartz inequality and Jensen's inequality(Lemma 18) imply

$$\int_{E_1} S_\epsilon(\mathbf{1}_{f \leq 0})(\mathbf{x}) - \mathbf{1}_{f(\mathbf{x}') \leq 0} d\gamma_1^*$$

$$\leq \left( \int_{E_1} \frac{1}{G\left(\phi(0) - C_\phi^*(\eta^*(\mathbf{x}'))\right)} d\gamma_1^* \right)^{\frac{1}{2}} \left( \int_{E_1} G\left(S_\epsilon(\phi \circ f)(\mathbf{x}) - \phi(f(\mathbf{x}'))\right) d\gamma_1^* \right)^{\frac{1}{2}} \tag{58}$$

$$\leq \left( \int \frac{1}{G\left(\phi(0) - C_\phi^*(\eta^*(\mathbf{x}'))\right)} d\gamma_1^* \right)^{\frac{1}{2}} G\left( \int_{E_1} S_\epsilon(\phi \circ f)(\mathbf{x}) - \phi(f(\mathbf{x}')) d\gamma_1^* \right)^{\frac{1}{2}},$$

which implies (55).

$\square$

# G   Technical Integral Lemmas

In this section, we require several technical facts about Riemann–Stieltjes integrals, which we briefly review here.

Let $g : \mathbb{R} \to \mathbb{R}$, $h : \mathbb{R} \to \mathbb{R}$ be functions and let $P = \{z_0, z_1, \ldots, z_K\}$ be a partition of an interval $I$. Then the *lower* and *upper* sums with respect to $g, h, P$ are defined as

$$L(g, h, P) = \sum_{k=0}^{K-1} \inf_{z \in [z_k, z_{k+1}]} g(z)(h(z_{k+1}) - h(z_k)), \quad U(g, h, P) = \sum_{k=0}^{K-1} \sup_{z \in [z_k, z_{k+1}]} g(z)(h(z_{k+1}) - h(z_k))$$

respectively. When $g$ is non-increasing, these simplify as $\inf_{z \in [z_k, z_{k+1}]} g(z) = g(z_{k+1})$ and $\sup_{z \in [z_k, z_{k+1}]} g(z) = g(z_k)$.

Riemann–Stieltjes integral $\int_I g dh$ can be approximated by upper and lower sums, much as in the classical Riemann case. The following result records the relevant approximation property:

**Proposition 3.** *Let $\int_I g dh$ be a Riemann-Stieltjes integral. If $g$ is continuous and $h$ is monotone, then the integral exists. Moreover, for any partition $P$, $L(g, h, P) \leq \int_I g dh \leq U(g, h, P)$. In addition, for any $\delta > 0$, there exists a partition $P$ for which $U(g, h, P) - \delta \leq \int_I g dh \leq L(g, h, P) + \delta$.*

For details, see Apostol (1974, Theorem 7.17) or Theorem 2.24 of Wheeden & Zygmund (1977) for the existence statement and Apostol (1974, Theorem 7.27) for a discussion of upper and lower integrals.

### G.1 The Lebesgue and Riemann–Stieltjes integral of an increasing function

The goal of this section is to prove (37), or namely:

**Proposition 4.** *Let $f$ be a non-increasing, non-negative, continuous function on an interval $[a, b]$ and let $\mathbb{Q}$ be a finite positive measure. Let $z$ be a random variable distributed according to $\mathbb{Q}$ and define $h(\alpha) = \mathbb{Q}(z \leq \alpha)$. Then*

$$\int_{(a,b]} f(z) d\mathbb{Q}(z) = \int_a^b f(\alpha) dh(\alpha)$$

*where the integral on the left is defined as the Lebesgue integral in terms of the measure $\mathbb{Q}$ while the integral on the right is defined as a Riemann–Stieltjes integral.*

*Proof.* Recall that when $f$ is monotonic, the Riemann-Stieltjes integral is the value of the limits

$$\int f dh = \lim_{\Delta \alpha_i \to 0} \sum_{i=0}^{I-1} f(\alpha_i)(h(\alpha_{i+1}) - h(\alpha_i)) = \lim_{\Delta \alpha_i \to 0} \sum_{i=0}^{I-1} f(\alpha_{i+1})(h(\alpha_{i+1}) - h(\alpha_i)), \tag{59}$$

where these limits are evaluated as the size of the partition $\Delta \alpha_i = \alpha_{i+1} - \alpha_i$ approaches 0 (Apostol, 1974, Exercise 7.3, Theorem 7.19), while the Lebesgue integral $\int f d\mathbb{Q}$ is defined as

$$\int f d\mathbb{Q} = \sup \left\{ \int g d\mathbb{Q} : g \leq f, g \text{ simple function, } \right\}.$$

The limits in (59) are upper and lower sums because $f$ is monotonic, and thus by Proposition 3, for any $\delta > 0$, one can choose a partition $\{\alpha_i\}_{i=0}^{I}$ for which each of the sums in (59) is within $\delta$ of $\int f dh$.

Next, consider two simple functions $g_1$, $g_2$ defined according to

$$g_1(z) = \sum_{i=0}^{I-1} f(\alpha_{i+1}) \chi_{z \in (\alpha_i, \alpha_{i+1}]}, \quad g_2(z) = \sum_{i=0}^{I-1} f(\alpha_i) \chi_{z \in (\alpha_i, \alpha_{i+1}]}.$$

By construction, $g_1(x) \leq f(x) \leq g_2(x)$ for all $x \in (a, b]$. Moreover, since $f(\alpha_i) - f(\alpha_{i+1}) < \delta$, it follows that $f(x) \leq g_2(x) + \delta$ when $x \in (a, b]$. Now applying the definition of the integral of a simple function, we obtain:

$$\int f dh - \delta \leq \sum_{i=0}^{I-1} f(\alpha_{i+1})\big(h(\alpha_{i+1}) - h(\alpha_i)\big) = \int_{(a,b]} g_1 d\mathbb{Q} \leq \int_{(a,b]} f d\mathbb{Q} \leq \int_{(a,b]} g_2 d\mathbb{Q}$$

$$= \sum_{i=0}^{I-1} f(\alpha_i)\big(h(\alpha_{i+1}) - h(\alpha_i)\big) \leq \int f dh + \delta$$

As $\delta$ is arbitrary, it follows that $\int f dh = \int f d\mathbb{Q}$. $\qquad\square$

Notice that because $H(0) = 0$, the integral in the right-hand side of (37) is technically an improper integral. Thus to show (37), one can conclude that

$$\int_{z \in (\delta, 1/2]} \frac{1}{H(z)} d\mathbb{Q}(z) = \int_\delta^{1/2} \frac{1}{H(\alpha)} dh(\alpha)$$

from Proposition 4 and then take the limit $\delta \to 0$.

### G.2 Proof of the last equality in (38)

The goal of this appendix is to prove the following inequality:

**Lemma 19.** *Let $h : [0, 1/2] \to [0, 1]$ be an increasing and right-continuous function with $h(0) = 0$ and $h(1/2) \leq 1$. Let $H$ be any continuous function with $H \geq h$ and let $r \in [0, 1)$. Then one can bound the Riemann–Stieltjes integral $\int 1/H(z)^r dh$ by*

$$\int_0^{1/2} \frac{1}{H(z)^r} dh \leq \frac{1}{1-r}$$

*Proof.* Let $\delta > 0$, then one can pick a partition $P = \{z_0 = 0, z_1, \ldots, z_K = 1/2\}$ for which $\int_0^{1/2} H^{-r} dh \leq L(H^{-r}, h, P) + \delta$. As $H^{-r}$ is non-increasing, $L(H^{-r}, h, P) = \sum_{k=0}^{K-1} H^{-r}(z_{k+1})(h(z_{k+1}) - h(z_k))$. Therefore, if we define $a_k = h(z_k)$, then

$$\int_0^{1/2} H^{-r} dh \leq \sum_{k=0}^{K-1} H^{-r}(z_{k+1})(h(z_{k+1}) - h(z_k)) + \delta \leq \sum_{k=1}^{K-1} h^{-r}(z_{k+1})(h(z_{k+1}) - h(z_k)) + \delta$$

$$= \sum_{k=0}^{K-1} a_{k+1}^{-r}(a_{k+1} - a_k) + \delta \tag{60}$$

Because the function $y \mapsto y^{-r}$ is decreasing in $y$, one can bound $a_{k+1}^{-r}(a_{k+1} - a_k) \leq \int_{a_k}^{a_{k+1}} y^{-r} dy$ and consequently the sum in (60) is bounded above as

$$\sum_{k=0}^{K-1} \int_{a_k}^{a_{k+1}} y^{-r} dy = \int_0^{h(1/2)} y^{-r} dy \leq \int_0^1 y^{-r} dy = \frac{1}{1-r}$$

Therefore $\int_0^{1/2} H^{-r} dh \leq 1/(1-r) + \delta$. The result follows as $\delta > 0$ is arbitrary. $\qquad\square$

## H Optimizing the Bound of Lemma 8 over $r$

*Proof of Theorem 11.* Let

$$f(r) = \frac{1}{1-r} a^r$$

Then

$$f'(r) = \frac{1}{(1-r)^2} a^r + \frac{1}{1-r} \ln a \, a^r$$

solving $f'(r^*) = 0$ produces $r^* = 1 + \frac{1}{\ln a}$, and

$$f\left(1 + \frac{1}{\ln a}\right) = -\ln a \, a^{1 + \frac{1}{\ln a}} = -ea \ln a$$

One can verify that this point is a minimum via the second derivative test:

$$f'(r) = \left(\frac{1}{1-r} + \ln a\right) f(r)$$

and thus

$$f''(r) = \left(\frac{1}{1-r} + \ln a\right) f'(r) + \frac{1}{(1-r)^2} f(r).$$

Consequently, $f''(r^*) = \ln(a)^2 f(1 + \frac{1}{\ln a}) > 0$.

However, the point $r^*$ is in the interval $[0, 1]$ only when $a \in [0, e^{-1}]$. When $a > e^{-1}$, $f$ is minimized over $[0, 1]$ at $r = 0$. Because $r^*$ is a minimizer when $a \in [0, e^{-1}]$, one can bound $f(0) \geq f(r^*)$ over this set and thus

$$f(r) \leq \min\left(1, -ea \ln a\right)$$

$$\square$$

# I Deferred proofs from Section 6

## I.1 Existence of Minimizers and Complementary slackness

The existence and complimentary slackness theorems of Appendix D extend to $R_\phi^\epsilon$. Observe that minimizers of $R_\phi$ may assume values in $\overline{\mathbb{R}}$; for example, with the exponential loss $\phi(\alpha) = e^{-\alpha}$ and the distribution defined by $\eta(\mathbf{x}) \equiv 1$, the unique minimizer of $R_\phi$ is $+\infty$. Just as in the non-adversarial scenario, $R_\phi^\epsilon$ may fail to attain its infimum over $\mathbb{R}$-valued functions. Nevertheless, Frank & Niles-Weed (2024a, Lemma 8) and Frank (2025, Theorem 6) guarantee the existence of a minimizer over $\overline{\mathbb{R}}$-valued functions.

**Theorem 16.** *Let $\phi$ satisfy Assumption 1. Then*

$$\inf_{f\ \mathbb{R}\text{-valued}} R_\phi^\epsilon(f) = \inf_{f\ \overline{\mathbb{R}}\text{-valued}} R_\phi^\epsilon(f).$$

*Furthermore, equality is attained at some Borel measurable, $\overline{\mathbb{R}}$-valued function $f^*$.*

Moreover, Theorem 7 of Frank & Niles-Weed (2024b) describes two conditions that characterize minimizers of $R_\phi^\epsilon$ and maximizers $\bar{R}_\phi$.

**Theorem 17** (Complementary Slackness). *The function $f^*$ minimizes $R_\phi^\epsilon$ and the measures $(\mathbb{P}_0^*, \mathbb{P}_1^*)$ maximize $\bar{R}_\phi$ over $\mathcal{B}_\epsilon^\infty(\mathbb{P}_0) \times \mathcal{B}_\epsilon^\infty(\mathbb{P}_1)$ iff the following two conditions hold:*

*1)*

$$\int S_\epsilon(\phi(f^*))d\mathbb{P}_1 = \int \phi(f^*)d\mathbb{P}_1^* \quad and \quad \int S_\epsilon(\phi(-f^*))d\mathbb{P}_0 = \int \phi(-f^*)d\mathbb{P}_0^*$$

*2)*

$$C_\phi(\eta^*, f^*) = C_\phi^*(\eta^*) \quad \mathbb{P}^*\text{-}a.e.$$

## I.2 Proof of Lemmas 9 and 10

As a preliminary step, we establish that if $\mathbb{P}_0^*$, $\mathbb{P}_1^*$ are induced by transport maps, then these maps determine the locations of maximizers of $\phi \circ f$ and $\phi \circ -f$.

**Lemma 20.** *Let $\mathbb{P}_0^*, \mathbb{P}_1^*$ be maximizers of $\bar{R}_\phi$ induced by the transport maps $T_0, T_1$ satisfying $\|T_0(\mathbf{x}) - \mathbf{x}\| \leq \epsilon$, $\|T_1(\mathbf{x}) - \mathbf{x}\| \leq \epsilon$. Then any minimizer $f^*$ of $R_\phi^\epsilon$ satisfies*

$$S_\epsilon(\phi \circ -f^*)(\mathbf{x}) = \phi(-f^*(T_0(\mathbf{x})) \quad \mathbb{P}_0\text{-}a.e. \quad (61) \qquad S_\epsilon(\phi \circ f^*)(\mathbf{x}) = \phi(f^*(T_1(\mathbf{x})) \quad \mathbb{P}_1\text{-}a.e. \quad (62)$$

*Proof.* We show (62), the argument for (61) is analogous.

Let $f^*$ minimize $R_\phi^\epsilon$; such a function exists by Theorem 16. The complementary slackness condition Item 1) in Theorem 17 yields

$$\int S_\epsilon(\phi \circ f^*)(\mathbf{x})d\mathbb{P}_1 = \int \phi \circ f^*(\mathbf{x}')d\mathbb{P}_1^* = \int \phi(f^*(T_1(\mathbf{x})))d\mathbb{P}$$

As the relation $\|T_1(\mathbf{x}) - \mathbf{x}\| \leq \epsilon$ implies $S_\epsilon(\phi \circ f^*)(\mathbf{x}) \geq \phi(f^*(T_1(\mathbf{x})))$ one can conclude (62). $\square$

Next, we verify strong duality for these restricted measures, utilizing the notation defined in Equations (39) and (40). The statement below implies Lemma 9.

**Lemma 21.** *Let $\mathbb{P}_0, \mathbb{P}_1, \mathbb{P}_0^*, \mathbb{P}_1^*, T_0, T_1, U_0, U_1$ and $Q$ be as in Lemma 9, and let $f^*$ minimize $R_\phi^\epsilon(\cdot; \mathbb{P}_0, \mathbb{P}_1)$. Then*

$$\mathbb{P}_0^*|_Q \in \mathcal{B}_\epsilon^\infty(\mathbb{P}_0|_{U_0}), \quad \mathbb{P}_1^*|_Q \in \mathcal{B}_\epsilon^\infty(\mathbb{P}_1|_{U_1}) \tag{63}$$

$$R_\phi^\epsilon(f^*; \mathbb{P}_0|_{U_0}, \mathbb{P}_1|_{U_1}) = \bar{R}_\phi(\mathbb{P}_0^*|_Q, \mathbb{P}_1^*|_Q).$$

*Consequently $f^*$ minimizes $R_\phi^\epsilon(\cdot; \mathbb{P}_0|_{U_0}, \mathbb{P}_1|_{U_1})$ while $\mathbb{P}_0^*|_Q, \mathbb{P}_1^*|_Q$ maximize $\bar{R}_\phi$ over $\mathcal{B}_\epsilon^\infty(\mathbb{P}_0|_{U_0}) \times \mathcal{B}_\epsilon^\infty(\mathbb{P}_1|_{U_1})$.*

*Proof.* By the definitions of $Q$, $U_0$, $U_1$

$$\mathbb{P}_0^*|_Q = \mathbb{P}_0|_{U_0} \sharp T_0, \quad \mathbb{P}_1^*|_Q = \mathbb{P}_1|_{U_1} \sharp T_1.$$

The relations $\|T_0(\mathbf{x}) - \mathbf{x}\| \le \epsilon$, $\|T_1(\mathbf{x}) - \mathbf{x}\| \le \epsilon$ imply (63). Next, let $\tilde{\eta} = d\mathbb{P}_1^*|_Q / d(\mathbb{P}_1^*|_Q + \mathbb{P}_0^*|_Q)$. On the set $Q$,

$$\tilde{\eta} = \eta^* \quad \mathbb{P}^*\text{-a.e.} \tag{64}$$

Next, let $f^*$ be a minimizer of $R_\phi^\epsilon(\cdot; \mathbb{P}_0, \mathbb{P}_1)$. Lemma 20 and the definitions of $T_0$, $T_1$ imply that

$$R_\phi^\epsilon(f^*; \mathbb{P}_0|_{U_0}, \mathbb{P}_1|_{U_1}) = \int S_\epsilon(\phi \circ f^*)(\mathbf{x}) \mathbf{1}_{U_1}(\mathbf{x}) d\mathbb{P}_1 + \int S_\epsilon(\phi \circ -f^*)(\mathbf{x}) \mathbf{1}_{U_0}(\mathbf{x}) d\mathbb{P}_0 \tag{65}$$

$$\int \phi(f^*(T_1(\mathbf{x}))) \mathbf{1}_Q(T_1(\mathbf{x})) d\mathbb{P}_1 + \int \phi(-f^*(T_0(\mathbf{x})) \mathbf{1}_Q(T_0(\mathbf{x})) d\mathbb{P}_0 \tag{66}$$

$$= \int \phi(f^*(\mathbf{x}')) \mathbf{1}_Q(\mathbf{x}') d\mathbb{P}_1 \sharp T_1 + \int \phi(-f^*(\mathbf{x}')) \mathbf{1}_Q(\mathbf{x}') d\mathbb{P}_0 \sharp T_0 \tag{67}$$

$$= \int C_\phi(\eta^*, f^*) \mathbf{1}_Q d\mathbb{P}^* \tag{68}$$

Since $f^*$ minimizes $R_\phi^\epsilon(f; \mathbb{P}_0, \mathbb{P}_1)$, the complimentary slackness condition Item 2) of Theorem 17 implies $C_\phi(\eta^*, f^*) = C_\phi^*(\eta^*)$. Equation 64 further implies $C_\phi^*(\eta^*) = C_\phi^*(\tilde{\eta})$ on $Q$ $\mathbb{P}$-a.e. and therefore,

$$\int C_\phi(\eta^*, f^*) \mathbf{1}_Q d\mathbb{P}^* = \int C_\phi^*(\tilde{\eta}) d\mathbb{P}^*|_Q = \bar{R}(\mathbb{P}_0^*|_Q, \mathbb{P}_1^*|_Q)$$

$\square$

These results show that restricted measures in the dual corresponds directly to restricted measures in the primal.

*Proof of Lemma 9.* Applying Theorem 7 to the restricted measures $\mathbb{P}_1|_{U_1}$, $\mathbb{P}_0|_{U_0}$ and invoking Lemma 21 yields the claim. $\square$

Finally, one can conclude Lemma 10 by comparing $R_\phi^\epsilon(f^*; \mathbb{P}_0|_{T_0^{-1}(Q^C)}, \mathbb{P}_1|_{T_1^{-1}(Q^C)})$ and $R_\phi^\epsilon(f^*; \mathbb{P}_0|_{U_0^C}, \mathbb{P}_1|_{U_1^C})$.

*Proof of Lemma 10.* Observe that $U_0^C = T_0^{-1}(Q^C)$, $U_1^C = T_1^{-1}(Q^C)$ and let $f^*$ be a minimizer of $R_\phi^\epsilon(\cdot; \mathbb{P}_0, \mathbb{P}_1)$. Then Lemmas 9 and 21 imply

$$R_{\phi,*}^\epsilon(\mathbb{P}_0|_{U_0}, \mathbb{P}_1|_{U_1}) = R_\phi^\epsilon(f^*; \mathbb{P}_0|_{U_0}, \mathbb{P}_1|_{U_1}) = \int_Q C_\phi^*(\eta^*) d\mathbb{P}^*$$

$$R_{\phi,*}^\epsilon(\mathbb{P}_0|_{U_0^C}, \mathbb{P}_1|_{U_1^C}) = R_\phi^\epsilon(f^*; \mathbb{P}_0|_{U_0^C}, \mathbb{P}_1|_{U_1^C}) = \int_{Q^C} C_\phi^*(\eta^*) d\mathbb{P}^* \tag{69}$$

Summing these:

$$R_\phi^\epsilon(f^*; \mathbb{P}_0|_{U_0}, \mathbb{P}_1|_{U_1}) + R_\phi^\epsilon(f^*; \mathbb{P}_0|_{\tilde{U}_0}, \mathbb{P}_1|_{\tilde{U}_1}) = \int C_\phi^*(\eta^*) d\mathbb{P}^* = R_{\phi,*}^\epsilon(f^*; \mathbb{P}_0, \mathbb{P}_1)$$

$\square$

## I.3  Proof of Theorem 10

*Proof of Theorem 10.* Let $Q = \{\mathbf{x}' : |\eta^*(\mathbf{x}') - 1/2| < \alpha\}$. Then Lemma 9 applied to $Q^C$ shows that $(\mathbb{P}_0^*|_{Q^C}, \mathbb{P}_1^*|_{Q^C})$ maximize $\bar{R}_\phi$ over $\mathcal{B}_\epsilon^\infty(\mathbb{P}_0|_{U_0}) \times \mathcal{B}_\epsilon^\infty(\mathbb{P}_1|_{U_1})$, with $U_0 = T_0^{-1}(Q^C)$ and $U_1 = T_1^{-1}(Q^C)$. Next, observe that simply scaling the inequality (14) shows that Theorem 9 applies even when $\mathbb{P}(\mathbb{R}^d) \le 1$. Consequently, Theorem 9 and Lemma 10 imply that

$$R^\epsilon(f; \mathbb{P}_0|_{U_0}, \mathbb{P}_1|_{U_1}) - R_*^\epsilon(\mathbb{P}_0|_{U_0}, \mathbb{P}_1|_{U_1}) \le \frac{1}{\phi(0) - C_\phi^*(1/2 - \alpha)} \left( R_\phi^\epsilon(f; \mathbb{P}_0|_{U_0}, \mathbb{P}_1|_{U_1}) - R_{\phi*}^\epsilon(\mathbb{P}_0|_{U_0}, \mathbb{P}_1|_{U_1}) \right)$$

$$\le \frac{1}{\phi(0) - C_\phi^*(1/2 - \alpha)} \left( R_\phi^\epsilon(f; \mathbb{P}_0, \mathbb{P}_1) - R_{\phi*}^\epsilon(\mathbb{P}_0, \mathbb{P}_1) \right)$$

Next, by Lemma 10, adding $R^\epsilon(f; \mathbb{P}_0|_{U_0^C}, \mathbb{P}_1|_{U_1^C}) - R_*^\epsilon(\mathbb{P}_0|_{U_0^C}, \mathbb{P}_1|_{U_1^C})$ to both sides of the inequality above results in

$$R^\epsilon(f; \mathbb{P}_0, \mathbb{P}_1) - R_*^\epsilon(\mathbb{P}_0, \mathbb{P}_1) \le$$
$$\frac{1}{\phi(0) - C_\phi^*(1/2 - \alpha)} \left( R_\phi^\epsilon(f; \mathbb{P}_0, \mathbb{P}_1) - R_{\phi*}^\epsilon(\mathbb{P}_0|_{U_0}, \mathbb{P}_1) \right) + R^\epsilon(f; \mathbb{P}_0|_{U_0^C}, \mathbb{P}_1|_{U_1^C}) - \int_Q C^*(\eta^*) d\mathbb{P}^*$$

The fact that $C^*(\eta^*) \ge 1/2 - \alpha$ on $Q$ while $S_\epsilon(\mathbf{1}_{f \le 0}) \le 1$, $S_\epsilon(\mathbf{1}_{f > 0}) \le 1$ implies that $R^\epsilon(f; \mathbb{P}_0|_{U_0^C}, \mathbb{P}_1|_{U_1^C}) - \int_Q C^*(\eta^*) d\mathbb{P}^* \le (\frac{1}{2} + \alpha)\mathbb{P}^*(|\eta - 1/2| < \alpha)$. Thus, the excess risk contribution from the region $A$ is at most $(1/2 + \alpha)\mathbb{P}^*(|\eta^* - 1/2| < \alpha)$. $\square$

# J  Further details from Examples 1 to 3

In Appendices J.2 and J.3, we use an operation analogous to $S_\epsilon$ that calculates the infimum of a function over an $\epsilon$-ball. Formally, we define:

$$I_\epsilon(g)(\mathbf{x}) = \inf_{\|\mathbf{x}' - \mathbf{x}\| \le \epsilon} g(\mathbf{x}'). \tag{70}$$

Next, we define a mapping $\alpha_\phi$ from $\eta \in [0, 1]$ to minimizers of $C_\phi(\eta, \cdot)$ by

$$\alpha_\phi(\eta) = \inf\{\alpha : \alpha \text{ is a minimizer of } C_\phi(\eta, \cdot)\}. \tag{71}$$

Lemma 25 of Frank & Niles-Weed (2024b) shows that the function $\alpha_\phi$ defined in (71) maps $\eta$ to the smallest minimizer of $C_\phi(\eta, \cdot)$ and is non-decreasing. This property will be instrumental in constructing minimizers for $R_\phi^\epsilon$.

## J.1  Proof of Lemma 2

*Proof of Lemma 2.* If $R_*^\epsilon = 0$, by Theorem 5, for any measures $\mathbb{P}_0' \in \mathcal{B}_\epsilon^\infty(\mathbb{P}_0)$, $\mathbb{P}_1' \in \mathcal{B}_\epsilon^\infty(\mathbb{P}_1)$ we have $\mathbb{P}'(\eta' = 0 \text{ or } 1) = 1$, where $\mathbb{P}' = \mathbb{P}_0' + \mathbb{P}_1'$ and $\eta' = d\mathbb{P}_1'/d\mathbb{P}'$. This statement must also hold for the $\mathbb{P}_0^* \in \mathcal{B}_\epsilon^\infty(\mathbb{P}_0)$, $\mathbb{P}_1^* \in \mathcal{B}_\epsilon^\infty(\mathbb{P}_1)$ that maximize $\bar{R}_\phi$. $\square$

## J.2  Calculating the optimal $\mathbb{P}_0^*$, $\mathbb{P}_1^*$ for Example 2

First, notice that a minimizer of $R_\phi$ is given by $f(x) = \alpha_\phi(\eta(x))$ with $\eta(x)$ as defined in (16). Below, we construct a minimizer $f^*$ for $R_\phi^\epsilon$. We'll do this construction separately for $\epsilon \le \delta$ and $\epsilon \in (\delta, 1 - \delta)$.

**When $\epsilon \le \delta$:**

Define a function $\tilde{\eta} : [-\delta - \epsilon - 1, 1 + \delta + \epsilon] \to [0, 1]$ by

$$\tilde{\eta}(x) = \begin{cases} \frac{1}{4} & \text{if } x \in [-1 - \delta - \epsilon, 0) \\ \frac{1}{2} & \text{if } x = 0 \\ \frac{3}{4} & \text{if } x \in (0, 1 + \delta + \epsilon] \end{cases}$$

and a function $f^*$ by $f^*(x) = \alpha_\phi(\tilde{\eta}(x))$.

We'll verify that this function is a minimizer by showing that $R_\phi^\epsilon(f^*) = R_\phi(f)$. As the minimal possible adversarial risk is bounded below by $R_{\phi,*}$, one can conclude that $f^*$ minimizes $R_\phi^\epsilon$. Consequently, $\bar{R}_\phi(\mathbb{P}_0, \mathbb{P}_1) = R_\phi^\epsilon(f)$ and thus the strong duality result in Theorem 7 would imply that $\mathbb{P}_0$, $\mathbb{P}_1$ must maximize the dual problem.

As both $\tilde{\eta}$ and $\alpha_\phi$ are non-decreasing, the function $f^*$ must be non-decreasing as well. Consequently, $S_\epsilon(\phi(f^*))(x) = \phi(I_\epsilon(f^*)(x)) = \phi(f^*(x-\epsilon))$ and similarly, $S_\epsilon(\phi(-f^*))(x) = \phi(-S_\epsilon(f^*)(x)) = \phi(-f^*(x+\epsilon))$. (Recall the $I_\epsilon$ operation was defined in (70).)

Therefore,

$$
R_\phi^\epsilon(f^*) = \int S_\epsilon(\phi(f^*))(x)p_1(x)dx + \int S_\epsilon(\phi(-f^*))(x)p_0(x)dx = \int \phi(f^*(x-\epsilon))p_1(x)dx + \int \phi(-f^*(x+\epsilon))p_0(x)dx
$$
$$
= \int \phi(f^*(x))p_1(x+\epsilon)dx + \int \phi(-f^*(x))p_0(x-\epsilon)dx
$$
(72)

Consequently,

$$
\int \phi(f^*(x))p_1(x+\epsilon)dx = \int_{-1-\delta-\epsilon}^{-\delta-\epsilon} \frac{1}{8}\phi\left(\alpha_\phi\left(\frac{1}{4}\right)\right)dx + \int_{\delta-\epsilon}^{1+\delta-\epsilon} \frac{1}{8}\phi\left(\alpha_\phi\left(\frac{3}{4}\right)\right)dx
$$
$$
= \int_{-1-\delta}^{-\delta} \frac{1}{8}\phi\left(\alpha_\phi\left(\frac{1}{4}\right)\right)dx + \int_{\delta}^{1+\delta} \frac{1}{8}\phi\left(\alpha_\phi\left(\frac{3}{4}\right)\right)dx = \int \phi(f(x))p_1(x)dx
$$

Analogously, one can show that

$$
\int \phi(-f^*(x))p_0(x-\epsilon)dx = \int \phi(-f(x))p_0(x)dx
$$

and consequently $R_\phi^\epsilon(f^*) = R_\phi(f)$.

**When** $\epsilon \in (\delta, 1+\delta)$**:**

We will show that $R_\phi^\epsilon(f^*) = \bar{R}_\phi(\mathbb{P}_0^*, \mathbb{P}_1^*)$ for the proposed attacks, proving that $\mathbb{P}_0^*$, $\mathbb{P}_1^*$ are dual optimal distributions. This time, define the function $\tilde{\eta} : [-\delta-\epsilon-1, 1+\delta+\epsilon] \to [0,1]$ by

$$
\tilde{\eta}(x) = \begin{cases}
0 & \text{if } x \in [-1-\delta-\epsilon, -1-\delta+\epsilon) \\
\frac{1}{4} & \text{if } x \in [-1-\delta+\epsilon, -(\epsilon-\delta)) \\
\frac{1}{2} & \text{if } x \in [-(\epsilon-\delta), (\epsilon-\delta)] \\
\frac{3}{4} & \text{if } x \in ((\epsilon-\delta), 1+\delta-\epsilon] \\
1 & \text{if } x \in (1+\delta-\epsilon, 1+\delta+\epsilon]
\end{cases}
$$

and again take $f^*(x) = \alpha_\phi(\tilde{\eta}(\mathbf{x}))$. The function $f^*$ is non-decreasing, so again (72) holds. Further, defining $p_1^*$, $p_0^*$ as $p_1^*(x) = p_1(x+\epsilon)$ and $p_0^*(x) = p_0(x-\epsilon)$ implies the relation

$$
R_\phi^\epsilon(f^*) = \int C_\phi(\eta^*, f^*)p^*(x)dx,
$$

where $p^*(x) = p_0^*(x) + p_1^*(x)$ and $\eta^* = p_1^*(x)/p^*(x)$. The function $\tilde{\eta}$ was defined so that $\tilde{\eta}(x) = \eta^*(x)$ a.e. and hence

$$
C_\phi(\eta^*, f^*) = C_\phi(\eta^*, \alpha_\phi(\eta^*)) = C_\phi^*(\eta^*).
$$

This relation implies $R_\phi^\epsilon(f^*) = \bar{R}_\phi(\mathbb{P}_0^*, \mathbb{P}_1^*)$, where $\mathbb{P}_0^*$, $\mathbb{P}_1^*$ are the distributions with pdfs $p_0^*$ and $p_1^*$.

### J.3 Calculating the optimal $\mathbb{P}_0^*$ and $\mathbb{P}_1^*$ for Example 3— Proof of (17)

We will show that the densities in (17) are dual optimal by finding a function $f^*$ for which $R_\phi^\epsilon(f^*) = \bar{R}_\phi(\mathbb{P}_0^*, \mathbb{P}_1^*)$. Theorem 7 will then imply that $\mathbb{P}_0^*$, $\mathbb{P}_1^*$ must maximize the dual. Define $\eta^*(x)$ by

$$\eta^*(x) = \frac{p_1^*(x)}{p_1^*(x) + p_0^*(x)},$$

with $p_0^*(x)$ and $p_1^*(x)$ as in (17). For a given loss $\phi$ we will prove that the optimal function $f^*$ is given by

$$f^*(x) = \alpha_\phi(\eta^*(x)).$$

The function $\eta^*$ computes to

$$\eta^*(x) = \frac{1}{1 + e^{\frac{\mu_1 - \mu_0 - 2\epsilon}{\sigma^2}(\frac{\mu_1 + \mu_0}{2} - x)}}.$$

If $\mu_1 - \mu_0 > 2\epsilon$, the conditional probability $\eta^*(x)$ is increasing in $x$ and consequently the function $f^*$ is non-decreasing. Therefore, $S_\epsilon(\phi(f^*))(x) = \phi(I_\epsilon(f^*)(x)) = \phi(f^*(x - \epsilon))$ (recall $I_\epsilon$ was defined in (70)). Similarly, one can argue that $S_\epsilon(\phi(-f^*))(x) = \phi(-f^*(x + \epsilon))$, and therefore,

$$R_\phi^\epsilon(f^*) = \int S_\epsilon(\phi(f^*))(x)p_1(x)dx + \int S_\epsilon(\phi(-f^*))(x)p_0(x)dx = \int \phi(f^*(x - \epsilon))p_1(x)dx + \int \phi(-f^*(x + \epsilon))p_0(x)dx$$

$$= \int \phi(f^*(x))p_1(x + \epsilon)dx + \int \phi(-f^*(x))p_0(x - \epsilon)dx.$$

Next, notice that $p_1(x + \epsilon) = p_1^*(x)$ and $p_0(x - \epsilon) = p_0^*(x)$. Define $\mathbb{P}^* = \mathbb{P}_0^* + \mathbb{P}_1^*$. Then

$$R_\phi^\epsilon(f^*) = \int \eta^* \phi(\alpha_\phi(\eta^*)) + (1 - \eta^*)\phi(-\alpha_\phi(\eta^*))d\mathbb{P}^* = \int C_\phi^*(\eta^*)d\mathbb{P}^* = \bar{R}_\phi(\mathbb{P}_0^*, \mathbb{P}_1^*)$$

Consequently, the strong duality result in Theorem 7 implies that $\mathbb{P}_0^*$ $\mathbb{P}_1^*$ must maximize the dual $\bar{R}_\phi$.

### J.4 Showing (18)

**Lemma 22.** *Consider an equal gaussian mixture with variance $\sigma$ and means $\mu_0 < \mu_1$, with pdfs given by*

$$p_0(x) = \frac{1}{2} \cdot \frac{1}{\sqrt{2\pi}\sigma}e^{-\frac{(x - \mu_0)^2}{2\sigma^2}}, \quad p_1(x) = \frac{1}{2} \cdot \frac{1}{\sqrt{2\pi}\sigma}e^{-\frac{(x - \mu_1)^2}{2\sigma^2}}$$

*Let $\eta(x) = p_1(x)/(p_0(x) + p_1(x))$. Then $|\eta(x) - 1/2| \leq z$ iff $x \in [\frac{\mu_0 + \mu_1}{2} - \Delta(z), \frac{\mu_0 + \mu_1}{2} + \Delta(z)]$, where $\Delta(z)$ is defined by*

$$\Delta(z) = \frac{\sigma^2}{\mu_1 - \mu_0} \ln\left(\frac{\frac{1}{2} + z}{\frac{1}{2} - z}\right). \tag{73}$$

*Proof.* The function $\eta$ can be rewritten as $\eta(x) = 1/(1 + p_0/p_1)$ while

$$\frac{p_0(x)}{p_1(x)} = \exp\left(-\frac{(x - \mu_0)^2}{2\sigma^2} + \frac{(x - \mu_1)^2}{2\sigma^2}\right) = \exp\left(\frac{\mu_1 - \mu_0}{\sigma^2}\left(\frac{\mu_1 + \mu_0}{2} - x\right)\right)$$

Consequently, $|\eta(x) - 1/2| \leq z$ is equivalent to

$$\frac{1}{2} - z \leq \frac{1}{\exp\left(\frac{\mu_1 - \mu_0}{\sigma^2}(\frac{\mu_1 + \mu_0}{2} - x)\right) + 1} \leq \frac{1}{2} + z$$

which is equivalent to

$$\frac{\mu_1 + \mu_0}{2} - \frac{\sigma^2}{\mu_1 - \mu_0} \ln\left(\frac{1}{\frac{1}{2} - z} - 1\right) \le x \le \frac{\mu_1 + \mu_0}{2} - \frac{\sigma^2}{\mu_1 - \mu_0} \ln\left(\frac{1}{z + \frac{1}{2}} - 1\right)$$

Finally, notice that

$$\Delta(z) = \frac{\sigma^2}{\mu_1 - \mu_0} \ln\left(\frac{1}{\frac{1}{2} - z} - 1\right) \tag{74}$$

while

$$-\Delta(z) = \frac{\sigma^2}{\mu_1 - \mu_0} \ln\left(\frac{1}{z + \frac{1}{2}} - 1\right)$$

$\square$

**Lemma 23.** *Let $p_0, p_1$, and $\eta$ be as in Lemma 22 and let $h(z) = \mathbb{P}(|\eta - 1/2| \le z)$. Then if $\mu_1 - \mu_0 \le \sqrt{2}\sigma$, then $h$ is concave.*

*Proof.* To start, we calculate the second derivative of $\Delta(z)$ and the first derivative of $p_0$.

The first derivative of $\Delta$ is

$$\Delta'(z) = \frac{\sigma^2}{\mu_1 - \mu_0} \cdot \frac{1}{\frac{1}{4} - z^2}.$$

and the second derivative of $\Delta(z)$ is

$$\Delta''(z) = \frac{\sigma^2}{\mu_1 - \mu_0} \cdot \frac{2z}{(\frac{1}{4} - z^2)^2} \tag{75}$$

Next, one can calculate the derivative of $p_0$ as

$$p_0'(x) = \frac{1}{2} \cdot \frac{1}{\sqrt{2\pi}\sigma} \cdot \frac{-(x - \mu_0)}{\sigma^2} e^{-\frac{(x-\mu_0)^2}{2\sigma^2}} = -\frac{(x - \mu_0)}{\sigma^2} p_0(x) \tag{76}$$

and similarly

$$p_1'(x) = -\frac{(x - \mu_1)}{\sigma^2} p_1(x) \tag{77}$$

Let $p(x) = p_0 + p_1$. Lemma 22 implies that the function $h$ is given by $h(z) = \int_{\frac{\mu_1 + \mu_0}{2} - \Delta(z)}^{\frac{\mu_1 + \mu_0}{2} + \Delta(z)} p(z)dz$. The first derivative of $h$ is then

$$h'(z) = \left(p\left(\frac{\mu_1 + \mu_0}{2} + \Delta(z)\right) + p\left(\frac{\mu_1 + \mu_0}{2} - \Delta(z)\right)\right)\Delta'(z).$$

Differentiating $h$ twice results in

$$h''(z) = \left(p\left(\frac{\mu_1 + \mu_0}{2} + \Delta(z)\right) + p\left(\frac{\mu_1 + \mu_0}{2} - \Delta(z)\right)\right)\Delta''(z)$$

$$+ \left(p'\left(\frac{\mu_1 + \mu_0}{2} + \Delta(z)\right) - p'\left(\frac{\mu_1 + \mu_0}{2} - \Delta(z)\right)\right)(\Delta'(z))^2$$

$$= \left(p\left(\frac{\mu_1 + \mu_0}{2} + \Delta(z)\right) + p\left(\frac{\mu_1 + \mu_0}{2} - \Delta(z)\right)\right)\left(\Delta''(z) - \frac{\Delta(z)\Delta'(z)^2}{\sigma^2}\right) \tag{78}$$

$$+ \left(p_1\left(\frac{\mu_1 + \mu_0}{2} + \Delta(z)\right) + p_1\left(\frac{\mu_1 + \mu_0}{2} - \Delta(z)\right)\right)\frac{\mu_1 - \mu_0}{2\sigma^2}(\Delta'(z))^2 \tag{79}$$

$$- \left(p_0\left(\frac{\mu_1 + \mu_0}{2} + \Delta(z)\right) + p_0\left(\frac{\mu_1 + \mu_0}{2} - \Delta(z)\right)\right)\frac{\mu_1 - \mu_0}{2\sigma^2}(\Delta'(z))^2. \tag{80}$$

where the final equality is a consequence of Equations (76) and (77). Next, we'll argue that the sum of the terms in Equations (79) and (80) is zero:

$$\left( p_1\left(\frac{\mu_1 + \mu_0}{2} + \Delta(z)\right) + p_1\left(\frac{\mu_1 + \mu_0}{2} - \Delta(z)\right)\right) - \left( p_0\left(\frac{\mu_1 + \mu_0}{2} + \Delta(z)\right) + p_0\left(\frac{\mu_1 + \mu_0}{2} - \Delta(z)\right)\right)$$

$$= \frac{1}{2\sqrt{2\pi}\sigma}\left(\left(e^{-\frac{\left(\frac{\mu_0 - \mu_1}{2} + \Delta(z)\right)^2}{2\sigma^2}} + e^{-\frac{\left(\frac{\mu_0 - \mu_1}{2} - \Delta(z)\right)^2}{2\sigma^2}}\right) - \left(e^{-\frac{\left(\frac{\mu_1 - \mu_0}{2} + \Delta(z)\right)^2}{2\sigma^2}} + e^{-\frac{\left(\frac{\mu_1 - \mu_0}{2} - \Delta(z)\right)^2}{2\sigma^2}}\right)\right)$$

$$= 0$$

Next, we'll show that under the assumption $\mu_1 - \mu_0 \leq \sqrt{2}\sigma$, the term (78) is always negative. Define $k = \sigma^2/(\mu_1 - \mu_0)$. Then

$$\Delta''(z) - \Delta(z)\frac{\Delta'(z)^2}{\sigma^2} = \frac{2k}{(\frac{1}{4} - z^2)^2}\left(z - \frac{k^2}{2\sigma^2}\ln\left(\frac{1}{\frac{1}{2} - z} - 1\right)\right) \tag{81}$$

The fact that $\Delta''(z) > 0$ for all $z$ implies that $\ln(1/(1/2 - z) - 1)$ is convex, and this function has derivative 4 at zero. Consequently, $\ln(1/(1/2 - z) - 1) \geq 4z$ and (81) implies

$$\Delta''(z) - \Delta(z)\frac{\Delta'(z)^2}{\sigma^2} \leq \frac{2k}{(\frac{1}{4} - z^2)^2}(z - \frac{k^2}{2\sigma^2} \cdot 4z) = \frac{2kz}{(\frac{1}{4} - z^2)^2}\left(1 - \frac{2k^2}{\sigma^2}\right)$$

The condition $\mu_1 - \mu_0 \leq \sqrt{2}\sigma$ is equivalent to $1 - 2k^2/\sigma^2 < 0$.

$\square$

This lemma implies that $h(z) \leq h'(0)z$. Noting also that $h(z) \leq 1$ for all $z$ produces the bound

$$h(z) \leq \min\left(\frac{16\sigma^2}{\mu_1 - \mu_0}z, 1\right)$$

applying this bound to the gaussians with densities $p_0^*$ and $p_1^*$ results in (18).