# OpenReview forum: "Adversarial Surrogate Risk Bounds for Binary Classification"
_TMLR — Accepted by TMLR_

### Review · Reviewer_ZJeZ · 2025-05-31

**Summary Of Contributions:**

This paper studies the consistency of surrogate loss in binary classification, especially in adversarial settings. The main claims are theoretical. It proves that (1) when Massart's noise condition is satisfied, a linear convergence bound can be achieved, and (2) a distribution-dependent bound may be achieved with mild conditions. Three toy examples are provided to illustrate different scenarios and assumptions.

**Audience:**

Yes

**Broader Impact Concerns:**

This is a theoretical work, and I do not see ethical concerns that require highlighting.

**Claims And Evidence:**

No

**Requested Changes:**

## Grammar and writing:
- Even the abstract does not fully parse: "Recent work characterized when a minimizing sequence of an adversarial surrogate risk is also a minimizing sequence of the adversarial classification risk for binary classification— a property known as adversarial consistency." I can guess the intention, but it is tough to parse and understand. Similar problems are all around. A global grammar check might be necessary.

## Assumption checking
- It is more proper to examine the restrictiveness of the assumptions in practical datasets like MNIST. I understand that a precise characterization is impossible, but numerical approximations should be provided.

## Result validation
- Providing numerical support for the established theorems increases confidence in the theories.

**Strengths And Weaknesses:**

Strength:
- This work provides important results regarding the surrogate loss consistency in adversarial settings.
- The theoretical results are extensive, characterizing different scenarios.
- Numerical examples are used to illustrate the idea.

Weakness:
- It does not highlight the difference of the adversarial setting and the standard setting, in terms of the theoretical tools and proof steps required. Concretely, this makes one wonder what is the main difference and key insights behind the proofs.
- No "slightly practical" examples (e.g., MNIST, CIFAR-10) are examined; the theoretical results are not backed by numerical simulations.

---

> ### Author Response · Authors · 2025-06-06
> **Technical Clarifications**
>
> # Reviewer ZJeZ
> ## Grammar
> We apologize that the grammar was difficult to read. To address this, we will use a large language model (LLM) to improve grammar and clarity before publication.
>
> ## Differences between the standard and the adversarial settings
> - **The challenge:** In the standard learning scenario, the excess risk $R(f)-R_{\ast}=\int C(\eta(\mathbf x),f(\mathbf x))-C^\ast(\eta(\mathbf x))d\mathbb P$ is the integral of a quantity that depends on each $\mathbf x \in\mathbb R^d$ independently. Surrogate risk bounds are typically established by deriving pointwise inequalities between the excess conditional risks $C(\eta(\mathbf x),f(\mathbf x)) -C^\ast(\eta(\mathbf x))$ and $C_\phi(\eta(\mathbf x),f(\mathbf x)) -C_\phi^\ast(\eta(\mathbf x))$ at each point $\mathbf x$ [1,2,3]. The crucial property enabling these bounds is that these excess conditional risks are *pointwise*--- they depend solely on the input $\mathbf x$. In contrast, in the adversarial setting, the excess risk $R^\epsilon(f)-R_\ast^\epsilon$ is not the integral of a pointwise quantity. As a result, the standard surrogate risk bounding techniques are not directly applicable.
> -  **Our approach:** The surrogate bounds presented in this work rely on a different strategy.  Specifically, Lemma 1 and the minimax theorems (Theorems 7 and 9) enable us to bound the excess risk in terms of pointwise conditional risks of the *distribution of optimal attacks*. This representation allows one to subsequently apply standard techniques to derive meaningful surrogate bounds, despite the non-pointwise nature of the adversarial risk.
>
>
> We will add this discussion to the 'Background' and 'Main Results' sections.
>
>
>
> ## References
> [1] Peter Bartlett, Michael Jordan, and Jon McAuliffe. " Convexity, Classification, and Risk Bounds." *Journal of the ASA*, 2006
>
> [2] Ingo Steinwart. "How to compare different loss functions and their risk". *Constructive Approximation*, 2007.
>
> [3] Tong Zhang. "Statistical behavior and consistency of Classification methods based on convex risk minimization". *The Annals of Statistics*. 2004.
>
> [4] Jylha, H. "The $\ell_\infty$ optimal transport: Infinite cyclical monotonicity and the existence of optimal transport maps." *Calculus of Variations and Partial Differential Equations", 2014.

---

> > ### Author Response · Authors · 2025-06-06
> > **Real World Datasets**
> >
> > ## Comparisons with real-world datasets:
> > - Experimental results from prior work suggest that, in real-world datasets, $\eta^\ast$ is typically concentrated near 0 and 1. [5] compute lower bounds on the adversarial classification risk for binary tasks, focusing on classifying digits '3' and '7' in MNIST under $\ell_2$ perturbations. Their lower bound remains close to 0 for $\epsilon \leq 3$ and increases to 0.2 at $\epsilon = 4$. Since $C^\ast(\eta^\ast)$ is maximized at $\eta^\ast = 1/2$, a low adversarial implies low mass of $|\eta^\ast - 1/2|$ near zero. Similar trends are observed on Fashion MNIST and CIFAR10. [6] extend these bounds to the multiclass setting, though extending adversarial surrogate bounds beyond binary classification remains an open problem.
> > - When the optimal adversarial risk is non-zero, the adversarial Bayes classifier may not be unique up to degeneracy. Even without adversarial perturbations, datasets like MNIST and CIFAR10 contain inherently ambiguous examples. [7] identify such cases—for instance, MNIST image 2597 lies between a '5' and a '3', while image 3558 is difficult to classify at all, despite being labeled as a '5'. In CIFAR10, image 6877 is ambiguous between a ship and an airplane, and image 2532 is similarly hard to identify. One would expect $\eta(x) = 1/2$ for such examples. [8] show that similar ambiguity arises in adversarial settings: under $\ell_\infty$ perturbations of size $8/255$, approximately 6% of adversarial examples are ambiguous. In the binary case, one would thus expect $\eta^\ast(x) = 1/2$ for these inputs. The uniqueness of the adversarial Bayes classifier in multiclass settings remains an open question.
> >
> >
> >
> >
> > ## References
> >
> > [5] Arjun Nitin Bhagoji, Daniel Cullina, and Prateek Mittal. "Lower bounds on Adversarial Robustness from Optimal Transport." *NeurIps.* 2019.
> >
> > [6] Characterizing the Optimal 0-1 Loss for a Multi-class Classification with a test-time Aattacker. *NeurIps*. 2023.
> >
> > [7] https://labelerrors.com/
> >
> > [8] Biran Bartoldson, James Diffenderfer, Konstantinous Parasyris, and Bhavya Kailkhura. "Adversariala Robustness Limits via Scaling-Law and Human-Alignment Studies". *ICML*. 2024.

---

> > > ### Author Response · Authors · 2025-06-06
> > > **Extending our bounds to real world datasets**
> > >
> > > ## When the Adversarial Bayes Classifier is not Unique
> > > - Our bounds remain informative in some restricted settings even when the adversarial Bayes classifier is not unique up to degeneracy. In particular:
> > >     >**Theorem**
> > >     >    Assume that there exist maximizers $\mathbb P_0^\ast$, $\mathbb P_1^\ast$ of $\bar R_\phi$ for which $\mathbb P^\ast(\eta^\ast=1/2)=\delta$ and $\mathbb P^\ast$ is induced by a transport map from $\mathbb P$. Let $\tilde \Phi$ be the function in Theorem 13, but with $H$ defined as $H(z)=\text{conc}(\mathbb P^\ast(0<|\eta^\ast-1/2|\leq z))$. Then
> > >     >    $$ R^\epsilon (f)-R_\ast^\epsilon\leq \tilde \Phi(R_\phi^\epsilon(f)-R^\epsilon_{\phi,\ast}) +\frac \delta 2 $$
> > >
> > > This result relies on structural properties of the optimal adversarial distribution. Prior work from optimal transport theory verifies the assumption on $\mathbb P^\ast$ under mild conditions: Theorem 3.5 of [4] states that whenever $\mathbb P$ is absolutely continuous with respect to Lebesgue measure and the norm $\|\cdot\|$ is strictly convex, the measure $\mathbb P^\ast$ is induced by a transport map. It's not clear whether this assumption is satisfies for common datasets like CIFAR10 and MNIST. Removing the assumption that $\mathbb{P}^\ast$ is induced by a transport map remains an open problem.
> > >
> > > The proof of the above result proceeds in three steps:
> > >
> > >   1. **Applicability of the surrogate bounds:** The surrogate bounds in this work apply whenever $\mathbb P_1(\mathbb R^d)+\mathbb P_0(\mathbb R^d)\leq 1$. (The only portion of the proofs that depends on the total mass of $\mathbb R^d$ was an application of a modified version of Jensen's inequality, see Lemma 12).
> > >
> > >   2. **Reduction via the transport map and restriction:** Let $T$ denote the transport map for which $\mathbb P^\ast=\mathbb P\sharp T$. Define sets
> > >     $$H=\{\mathbf x: \eta^\ast(T(\mathbf x))=1/2\},\quad Q=\{\mathbf x: \eta^\ast(\mathbf x)=1/2\}.$$
> > >     By applying the minimax theorems, one can show that when the original distributions are restricted to the complement of the ambiguous region $S=H^C$ -- i.e., to the measures $\mathbb{P}_0\vert_S$ and $\mathbb{P}_1\vert_S$ -- the corresponding maximizers of the dual problem are given by the restrictions $\mathbb{P}_0^\ast\vert_U$
> > > and $\mathbb{P}_1^\ast\vert_U$, with $U=Q^C$. As a result, Theorem 13 applies directly to the restricted measures $\mathbb P_0\vert_S$, $\mathbb P_1\vert_U$.
> > >
> > >   3. **Combining errors inside and outside the ambiguous region:** The above argument provides a surrogate risk bound outside the ambiguous set $H$. Within $H$, the classifier $f$ incurs an error of at most $\delta$, while the Bayes optimal error is $\delta/2$. Accounting for the error over $H$ yields the final bound as stated in the theorem.
> > >
> > > The analogous result for Theorem 12 reads as follows:
> > >
> > >    >**Theorem**:
> > >    >   Assume that there exist maximizers $\mathbb P_0^\ast$, $\mathbb P_1^\ast$ of $\bar  R_\phi$ for which $\mathbb P^\ast(\eta^\ast=1/2)=\delta$ and there exists a constant $\alpha$ for which $\mathbb P^\ast(|\eta^\ast-1/2|\in(0,\alpha))=0$. Further assume that $\mathbb P^\ast$ is induced by a transport map from $\mathbb P$. Then
> > > $$R^\epsilon(f)-R_\ast^\epsilon\leq \frac{3+\sqrt 5}2 \frac 1 {\phi(0)-C_\phi^\ast(1/2-\alpha)} (R_\phi^\epsilon(f)-R_{\phi,\ast}^\epsilon)+\frac \delta 2.$$
> > >
> > >
> > > An instance where this result is applicable is Example 2 in the paper, particularly when $ \delta<\epsilon<1$. We will expand the discussion of Example 2 to address this regime as well.
> > >
> > > We will include this discussion in the final version of our paper. We will also include a figure demonstrating ambiguous images on MNIST and CIFAR10.

---

### Review · Reviewer_UYjp · 2025-07-21

**Summary Of Contributions:**

For the binary classification tasks, the authors extend the existing work of (Barlett et al., 2006) and provide surrogate risk bounds in the presence of adversarial perturbations. Firstly, the authors provide a linear risk bound under Massart's noise condition, with the convergence risk quantified. The authors also give a realizable risk bound when the adversarial Bayes risk is zero and a data-distribution dependent risk bound when the adversarial Bayes classifier is unique up to degeneracy. Moreover, the authors provide three examples where the results are illustrated.

**Audience:**

Yes

**Broader Impact Concerns:**

no concern

**Claims And Evidence:**

Yes

**Requested Changes:**

See weakness for details:
(1) and (3) should be addressed to some extent
(2) is critical
(4) is optimal

**Strengths And Weaknesses:**

Strengths:

(1) The authors extend the result of (Barlett et al., 2006) to the adversarial setting, which is of broad interest.
(2) The technical parts appear correct.

Weakness:
(1) The bound in Theorem 11 is not optimized; therefore, its tightness is neither analyzed nor demonstrated experimentally.
(2) The writing of section 2 can be improved. It is difficult to see which are existing works and which are the proposals (directions) of this work, and how they are connected. Since the adversarial setting and proofs of the main results rely on the works (e.g. Frank&Niles-Weed,2024a,2024b), the clarity of Section 2 is important.
(3) The Massart's noise condition is difficult to verify in practice. Can the authors give more interpretation or demonstrate the applicability in toy experiments (e.g. MNIST)?
(4) The proof of Theorems 11 and 12 can be shortened by focusing on the calibration of the space into D.E.F.

Minor:
A repetition in section 6 (...tool central tool..)

---

> ### Author Response · Authors · 2025-07-28
>
> ## (1)
> - For the hinge loss $\phi(z)=(1-z)_+$, the constant in Theorem 11 is at most $(3+\sqrt 5)/2$ larger than the optimal value. Consider a distribution on $[-\epsilon,\epsilon]$ for which $\mathbb P^*(\eta=1/2+\alpha)=1$ and let $\phi(z)=(1-z)_+$ be the hinge loss. Consider the sequence of functions given in (14) of the paper. Then $R^\epsilon(f_n)-R^\epsilon_*=1/2+\alpha$ while $\lim_{n\to \infty} R_\phi^\epsilon(f_n)=\phi(0)-C_\phi^*(1/2+\alpha)$. Consequently,
> $$ \lim_{n\to \infty} \frac{R^\epsilon(f_n)-R^\epsilon_{\phi,*}}{R^\epsilon(f_n)-R^\epsilon_*} =\frac{\frac 12 +\alpha}{\phi(0)-C_\phi^*(1/2-\alpha)}\leq \frac 1 {\phi(0)-C_\phi^*(1/2-\alpha)}$$
> - A simple example show that the similar bound in Theorem 10 is tight: Consider a distribution for which $\mathbb P(\eta=0)=1$ and $f(\mathbf x)\equiv 0$.
> - The bound in Theorem 11 is not tight for all loss functions. Prior work shows that $$ R^\epsilon(f)-R^\epsilon_*\leq  \frac 1 {2(\phi(0)-C_\phi^*(1/2))} R^\epsilon_\phi(f)-R^\epsilon_{\phi,*}$$ specifically for the $\rho$-margin loss. Whether there exists a loss for which Theorem 11 is tight remains an open question.
> -We will add this discussion to the paper
>
> ## (2)
> - Theorems 1-2, 4-9, 10 part 2 and  Lemma 1 have all appeared in prior work
> - Theorem 3, Proposition 1, and Theorem 10 (part 1) are, to our knowledge, new—though not particularly novel. For example, a weaker version of Proposition 1 follows from the surrogate bound in [1] when $C_\phi^*(1/2) < \phi(0)$. These results are included primarily because they are required for the proofs of the main theorems.
> - To remedy this confusion, we will make sure to clearly cite prior work and add the citation to the theorem header for Theorems 1-2, 4-9. We will also explicitly indicate which results (Theorem 3, Proposition 1, Theorem 10 part 1) are new.
>
>
> ## (3)
> Some prior work has analyzed properties of data distributions under adversarial context. Comparing their conclusions with our assumptions is insightful. Reviewer ZJeZ raised a similar point, which we addressed in the comments titled "Comparisons with real-world datasets" and "Extending our bounds to real-world datasets." Please confirm that you are able to view those comments.
>
>
> ## (4)
> We don't understand this comment. We would welcome suggestions on how to shorten the proofs. Could you please clarify? Also, what does "D.E.F." refer to?
>
>
>
> [1] Peter Bartlett, Michael Jordan, and Jon McAuliffe. " Convexity, Classification, and Risk Bounds." *Journal of the ASA*, 2006

---

> > ### Author Response · Authors · 2025-07-28
> > **(1) properly rendered**
> >
> > Our apologies--- (1) did not render properly in the last comment. Here is (1) again:
> > ## (1)
> > - For the hinge loss $ \phi(z)={(1-z)}_+ $, the constant in Theorem 11 is at most $ (3+\sqrt 5)/2 $ larger than the optimal value. Consider a distribution on $ [-\epsilon,\epsilon] $ for which $ {\mathbb P^\ast(\eta=1/2+\alpha)=1} $.
> >
> >  Consider the sequence of functions given in (14) of the paper. Then $ R^\epsilon(f_n)-R^\epsilon_\ast=1/2+\alpha $ while $ \lim_{n\to \infty} R_\phi^\epsilon(f_n)=\phi(0)-C_\phi^\ast(1/2+\alpha) $. Consequently,
> > $$ \lim_{n\to \infty} \frac{R^\epsilon(f_n)-R^\epsilon_{\phi,\ast}}{R^\epsilon(f_n)-R^\epsilon_\ast} =\frac{\frac 12 +\alpha}{\phi(0)-C_\phi^\ast(1/2-\alpha)}\leq \frac 1 {\phi(0)-C_\phi^\ast(1/2-\alpha)} $$
> > - A simple example show that the similar bound in Theorem 10 is tight: Consider a distribution for which $\mathbb P(\eta=0)=1$ and $f(\mathbf x)\equiv 0$.
> > - The bound in Theorem 11 is not tight for all loss functions. Prior work shows that $$ R^\epsilon(f)-R^\epsilon_\ast\leq  \frac 1 {2(\phi(0)-C_\phi^\ast(1/2))} R^\epsilon_\phi(f)-R^\epsilon_{\phi,\ast}$$ specifically for the $\rho$-margin loss. Whether there exists a loss for which Theorem 11 is tight remains an open question.
> > - The constant $(3+\sqrt 5)/2$ in Theorem 11 is the best one can achieve using our proof technique.

---

> > ### Comment · Reviewer_UYjp · 2025-08-05
> > **Clarification to (4)**
> >
> > By D,E,F, I am refering to equations (28) - (30) where you defined the sets D_1, E_1, F_1.

---

> > > ### Comment · Reviewer_UYjp · 2025-08-05
> > > **Proof shorten suggestions**
> > >
> > > To provide a clearer passage in the main paper, consider moving some intermediate steps to the appendix and citing them in the main content. I suggest, for example, moving the lines around equations (22)-(23) to the appendix and only leaving the resulting (22) as a lemma in the main part.

---

> > > > ### Author Response · Authors · 2025-08-05
> > > >
> > > > Thank you for the suggestion.
> > > >
> > > > We will separate equations (22) and (23) into a standalone lemma and move the detailed argument to the appendix.
> > > >
> > > > For Theorem 11, we will revise both the proof and the accompanying exposition to better highlight the distinct bounds on $D_1$, $D_0$, $E_1$, $E_0$, and $F_1$, $F_0$. We will also include an additional lemma to improve clarity.

---

### Review · Reviewer_XyY9 · 2025-07-29

**Summary Of Contributions:**

This paper studies the adversarial surrogate risk bounds for binary classification. The authors prove the linear convergence rates when the distribution of optimal adversarial attacks satisfies Massart's noise condition.

**Audience:**

Yes

**Claims And Evidence:**

Yes

**Requested Changes:**

I appreciate it if the authors can:

1, Move some redundant content to the appendix to facilitate the readers' understanding.

2. Write the references of the theorems clearly within the theorems.

3. Add some references to the introduction of the statistics background, for example, the second paragraph of the paper.

4: Do the grammar check for the sentence: To the best of the authors’ knowledge this paper is the first to prove surrogate risk bounds... on Page 2.

5. Make the labels and references of the equations consistent: Equation 8 is a consequence of (7) ...

6. Clarify if Theorem 3 is inspired by Theorem 2 in (Frank & Niles-Weed, 2024a).

7. Instead of listing the proofs for the main results, state the technical challenges/insights/differences of proving the surrogate risk bounds for the adversarial binary classification, compared with the non-adversarial setting.

**Strengths And Weaknesses:**

Strengths:

1. As previous work does not derive the adversarial risk bounds, the results in this paper fill this gap.
2. This paper provides detailed and rigorous proofs.

Weakness:

1. The organization of this paper can be improved. The body of the paper is a mix of proof and statement. Also, for example, Section 2.1, which is not the main part of this paper, occupies almost three pages. Also, it is hard to distinguish between the theorems/propositions cited from the existing literature and the ones newly proposed in this paper.

2. Massart's noise condition seems not general enough.

3. As the authors already mention, the optimality of the bound and the dependence on the sample size are not investigated, which limits the contributions of this paper.

---

> ### Author Response · Authors · 2025-08-01
> **Proposed edits**
>
> ## (0) Optimality of bound
> Per the request of another reviewer we can now show that Theorem 10 is tight while the constant in Theorem 11 is at most $(3+\sqrt 5)/2$ larger than the optimal value.
>
>
> ## (1)
> - We are working on shortening Section 2. Our preliminary revision plan includes the following changes:
>   - Split Section 2.1 into two subsections—**Surrogate Risk** and **Surrogate Risk Bounds**—to improve organization.
>   - Move the last two paragraphs on page 4 to the Related Works section.
>   - Remove the discussion of equation (14) from Section 2.
>   - Remove the introduction of equations (9) and (10) prior to Proposition 1.
>   - Consolidate the definitions of adversarial classification risk and adversarial surrogate risk, and streamline the surrounding exposition.
>   - The existence of an $\overline{\mathbb{R}}$-valued minimizer in Theorem 9 is not used until the appendix; we will relocate this discussion accordingly.
>
>
> ## (2)
> We will make sure to clearly cite prior work and add the citation to the theorem header for Theorems 1-2, 4-9. We will also explicitly indicate which results of Section 2 (Theorem 3, Proposition 1, Theorem 10 part 1) are new.
>
> ## (3)
> We cited the most pertinent results in the third paragraph of the introduction. Please let us know if we are missing major references.
>
> ## (4)
> Here is a clearer version:
>
> "To the best of our knowledge, this work presents the first surrogate risk bounds for the loss functions most commonly employed in adversarial training. A detailed comparison with prior work is provided in \cref{sec:related_works}."
>
> ## (5)
> We will make sure these are consistent in the final version of the paper
>
> ## (6)
> - It was actually inspired by Proposition 1 and Theorem 11, along with the proof of Theorem 13. Theorem 3 implies that the bounds in Proposition 1 and Theorem 11 do not divide by zero, which is also a key observation for the proof of Theorem 13.
> - However, Theorem 3 implies every adversarially consistent loss is also consistent. We will add this observation to the paper.
>
>
> ## (7)
> - In the final version of the paper, we will clarify how the adversarial setting differs from the standard (non-adversarial) case and provide a high-level overview of the main technical challenges involved in our proofs.
> - We have included only the key proofs in the main text. Given the theoretical nature of the paper, we believe this focus is appropriate and helps maintain clarity.

---

> > ### Comment · Reviewer_XyY9 · 2025-08-05
> >
> > Could the authors submit a revised version to Openreview?

---

### Decision · Action_Editor_3vvx · 2025-10-03

**Recommendation:** Accept with minor revision

**Additional Comments:**

Here are assorted comments from AC's side, after taking a look at the last version.

- Examples of loss functions should be mentioned here and there, for example, for Proposition 1 and Theorem 9. The condition $\\phi(0)>C\_\\phi\^\*(1/2-\\alpha)$ in Proposition 1 is not satisfied by most of the common surrogate losses (such as the logistic and hinge losses), so giving an example is important for standard readers. The $\\rho$-margin and shifted sigmoid losses seem to satisfy this.
- If you use a citation as a noun in a sentence, \citet is appropriate. For example, right before Eq. (1), " (Frank & Niles-Weed, 2024a, Proposition 3) show that ..." should be "Frank & Niles-Weed (2024a, Proposition 3) show that ..."
- Section 4 should presumably be titled with "proof" and Section 4.3 be discussed somewhere else.
- [The discussion on the bound tightness](https://openreview.net/forum?id=Bay1cHLk7h&noteId=szsjxkzDhk) can be expanded in the paper.

I recommend to address them when submitting the final version.

**Audience:**

Yes

**Audience Explanation:**

In the last couple of years, the theory community of adversarial robustness has been concerned with the consistency, and gave a basic characterization of the necessary conditions. Yet, they tend to overly pessimistic while real-world distributions are not that "pathological." Here is the motivation of this work to consider the adversarial consistency *unique up to degeneracy*. By complementing to the consistency characterization of Frank (2025), this work establishes a linear surrogate risk bound, which solves a theoretically challenging question left in the community. Overall, this is relevant to the community's interest.

**Claims And Evidence:**

Yes

**Claims Explanation:**

The reviewers confirmed the correctness of the theoretical claims. After the author-reviewer discussions, the authors thoroughly revised the paper to incorporate the reviewer's feedback, and the reviewers acknowledge the improved standard of the clarity.

---

> ### Author Response · Authors · 2025-10-07
>
> Thanks for your feedback. We made the following changes to the final version:
>
> - Proposition 1 and Theorem 9 apply for all consistent losses so long as $\alpha>0$. We made sure to emphasize this point in the paper. We also further emphasized which losses satisfy $C_\phi^*(1/2)<\phi(0)$
> - Citation as noun: Thanks, we fixed this grammar issue.
> - We updated the title of section 4
> - We expanded the discussion of the lower bound both in the "Main Results" section and section 4.3

---

> > ### Comment · Action_Editor_3vvx · 2025-10-08
> >
> > > Citation as noun: Thanks, we fixed this grammar issue.
> >
> > The same issue still persists in other places. After a quick glance,
> >
> > - At the two lines after Theorem 1: "(Frank & Niles-Weed, 2024a, Proposition 3) states [...]" -> "Frank & Niles-Weed (2024a, Proposition 3) states [...]"
> > - At the one line before Theorem 2: "Specifically, Bartlett et al. (2006) show" -> "Specifically, Bartlett et al. (2006) show[s]"
> > - The theorem header of Theorem 2: "Theorem 2 ((Tewari & Bartlett, 2007))" -> "Theorem 2 (Tewari & Bartlett, 2007)" by \citet
> > - Right after Definition 2: "Theorem 2 of (Frank & Niles-Weed, 2024a) characterizes" -> "Frank & Niles-Weed (2024a, Theorem 2) characterizes"
> >
> > I would not make the exhaustive list beyond this.
> >
> > Another (super minor) issue: in eq. (1), use `\left( ... \right)` to make the parentheses look nicer.
> >
> > The others look good.